# Short and long-term costs of inbreeding in the lifelong-partnership in a termite

Pierre-André Eyer [1 ✉] & Edward L. Vargo[1]

Social life and lifelong partner commitments are expected to favor thorough partner choice, as an ill-suited partnership may have long-term consequences, adversely affecting the parents and spanning several cohorts of offspring. Here, we used ~1400 termite incipient colonies to estimate the short- and long-term costs of inbreeding upon the survival of the parents over a 15-month period, their productivity, and the resistance of their offspring toward pathogen pressure. We observed that foundation success was not influenced by the relatedness of partners, but by their levels of microbial load. We showed faster growth in inbred colonies with low levels of microbial load, revealing a potential tradeoff between pathogen defense and offspring production. Yet, inbreeding takes its toll later in colony development when offspring from incipient colonies face pathogen pressure. Although the success of a lifetime partnership is initially determined by the partner's health, the cost of inbreeding in incipient colonies favors outbred colonies reaching maturity.

[1] Department of Entomology, 2143 TAMU, Texas A&M University, College Station, TX 77843-2143, USA. ✉email: pieyer@live.fr

The difference between the sexes in their gamete and off-spring investment generally leads to females being considered the choosy sex and males the more promiscuous sex. However, in high fidelity species, epitomized by the social Hymenoptera where males live as stored sperm, a detrimental mating cannot be remedied by new reproductive events. Lifelong partner commitments are expected to favor extreme choosiness by both sexes[1,2]. Additionally, the consequences of poor mate choice are higher for social species as the parents may be adversely affected, since they rely on their offspring for care, not only for themselves but also for rearing their future brood. Therefore, an ill-suited partnership may have long-term consequences, spanning several cohorts of offspring.

Mating with close relatives is commonly seen as detrimental due to the deleterious consequences of inbreeding (i.e., inbreeding depression), which logically suggests that evolution favors mechanisms preventing its occurrence[3]. Particularly well-studied in social and/or monogamous groups, inbreeding avoidance may arise through increased dispersal, reducing the likelihood of encountering relatives[4], or through delayed reproduction via parental inhibition, preventing mating between the parents and their offspring[5,6]. Remarkably, this sexual repression is lost when the opposite-sex parent is absent or replaced[7,8]. Inbreeding may also be reduced through extra-group fertilizations, whereby offspring are not fathered by the males in their group, despite caring for the offspring[9–11]. In some species, the highly synchronized swarming of a large number of reproducing individuals may reduce inbreeding by decreasing the chance of mating with a relative[12]. Finally, inbreeding avoidance may occur through recognition and avoidance of kin matings[13–15]. In some cases, the scent of males is unattractive and may even inhibit sexual behavior in their female relatives[16].

Termites are diplo-diploid eusocial insects that usually establish their colonies through the pairing of a winged queen and an unrelated king (i.e., outbreeding)[17]. The royal couple spends their entire lives together secluded within the colony, therefore usually preventing extra-pair fertilizations (colony fusion may allow extra-pair fertilizations in some cases). During colony foundation, the queen and king frequently engage in social interactions, such as grooming and trophallactic exchanges[18], and founding success is directly tied to the health of each partner[19]. The absence of workers prevents founding colonies from reaping the full benefits of social immunity, as workers collectively enhance disease resistance through the maintenance of nest hygiene, allogrooming, and the exchange of antimicrobial substances[20–22]. In incipient colonies, the parents' limited resources are drained by the production and care of the first brood, which is altricial for the two first instars and potentially more susceptible to pathogens than older workers[19,23,24]. The success of incipient colonies, therefore, increases with the body size of the founders and their contribution to biparental care[19,25,26]. However, as the colony grows, brood care, food foraging, and immune maintenance are undertaken by older workers, whereas the queen and king forego their parental duties to specialize in reproduction[27]. These behavioral and physiological changes highlight that, in addition to its requirement for mating, the presence of both partners and their mutual compatibility plays an important role in influencing the success of incipient colonies. They also emphasize the changing roles queens and kings play within colonies, questioning whether these different pressures influence selection for distinct partner traits over the lifespan of a colony.

Several lines of evidence suggest that inbreeding hampers the development of termite colonies. In *Zootermopsis angusticollis*, inbred groups are more susceptible to a fungal pathogen and exhibit higher cuticular microbial loads, potentially resulting from less-effective allogrooming[28]. In *Reticulitermes flavipes*, a high proportion of reproductives pair up with nestmates during the nuptial flight (25%); yet this proportion is reduced among established colonies, suggesting that inbreeding negatively affects colony development[29]. However, the susceptibility of mature colonies of *R. flavipes* toward pathogens has not been found to be associated with their level of inbreeding[30]; rather, specific genetic backgrounds seem to determine their survival to a greater extent than overall genetic diversity. Similarly, increased diversity from colony fusion in this species was not found to improve survival toward pathogens. Merged colony survival was instead equal to that of either the more susceptible or the more resistant colony, highlighting the complementary roles of both colonies of origin[31]. Similarly, inbreeding does not seem detrimental during colony establishment in *Z. angusticollis,* and offspring production was reported to be similar between inbred and non-inbred pairs of reproductives. However, the survival of incipient colonies was remarkably higher when initiated by inbred reproductives, which the authors suggested likely resulted from the immune priming of nestmate reproductives toward *familiar* pathogens due to prior exposure within their natal colony[32]. In contrast, high mortality in outbred pairings in *Z. angusticollis* may stem from non-nestmates facing *naïve* pathogens carried by their partner, toward which they may be more vulnerable[32].

Here, we sought to untangle the complex interaction between inbreeding and pathogen pressure on colony foundation in termites. Using six stock colonies of *R. flavipes*[33,34], we set up inbred and outbred pairings. We first investigated the short-term cost of outbreeding by assessing the influence of genetic relatedness, microbial loads, and microbial similarities on the foundation success of ~800 established pairings over the first 14 days. Second, we used ~1400 pairings to investigate the long-term cost of inbreeding by comparing inbred and outbred pairings over a 15-month period for their survival, their productivity (worker and soldier), and the susceptibility of their offspring toward entomopathogenic pressure. Overall, we show that inbreeding and outbreeding entail different costs at distinct stages of a colony's lifespan; identifying those costs can shed light on the evolutionary pressures influencing partner choice and inbreeding avoidance.

## Results

**Short-term survival of alate pairings**. To investigate the short-term effect of inbreeding on founding success, we set up inbred colonies established from sib alate pairings and outbred colonies from pairings between alates from different stock colonies for every combination of colonies. Fourteen days after pairing, only 101 incipient colonies (202 alates) of the 831 established pairings survived; 35 out of the 231 inbred pairings (15.15%) and 66 out of the 600 outbred pairings (11.00%). No significant difference was observed between the survival of inbred and outbred pairings ($P = 0.212$; Fig. 1a). However, strong differences in survival were observed between specific pairings ($P < 0.001$), ranging from a 47.5% survival for pairing AA to complete mortality for pairings AE and EE (the survival curve of each pairing is provided in Fig. S2). Alates from colony A had the highest survival rate, with 74 out of the 202 surviving alates originating from this colony (Fig. 2a). Pairings including an alate from A showed good survival overall (low hazard ratio), with the best survival observed for the inbred AA combination (Fig. 2b). Notably, the opposite was also observed, with alates from colony E having the highest mortality rate. Consequently, pairings including an alate from this colony had low survival, with the lowest survival observed for the inbred pairing EE (Fig. 2a, b). Overall, these results suggest that inbreeding has no effect on pairing survival in the first

### a. Short-term survival of inbred and outbred pairings

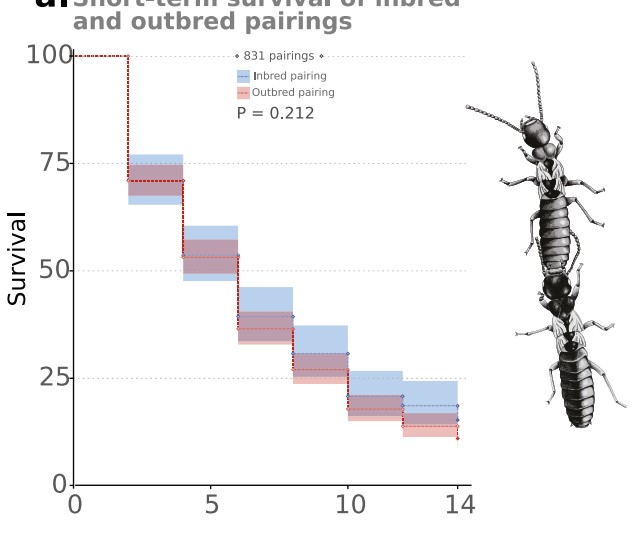

### b. Long-term survival of inbred and outbred pairings

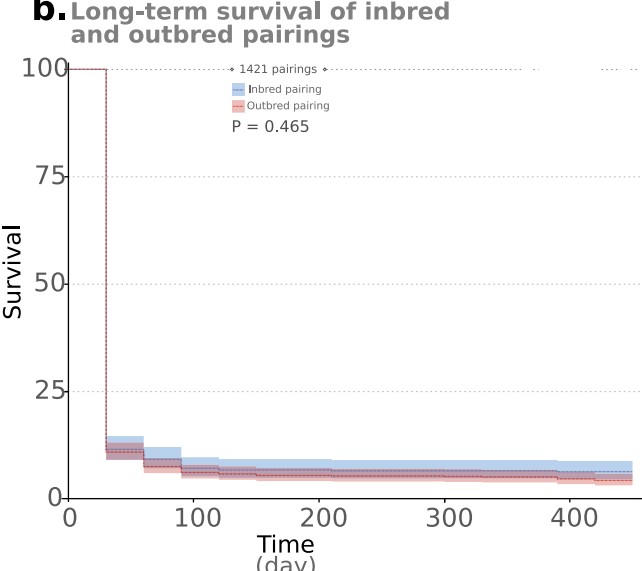

**Fig. 1 Short and long-term survival of inbred and outbred pairings.**
Kaplan–Meier survival distributions of inbred and outbred incipient colonies during the first 14 days after pairing (**a**) and along the overall length of the experiment (450 days; **b**).

several days after mating; rather, the survival of the pairings is strongly influenced by the colony of origin of the constituent partners.

The six colonies varied in their microbial loads obtained from the number of colony-forming units (CFUs) cultured from individual cuticular washes, with colonies A, D, and F exhibiting few CFUs (0.36, 0.39, and 0.58 for A, D, and F, respectively; Fig. 2d). In comparison, colonies B and C (and E to a lesser extent) displayed higher levels of microbial load with 14.86, 12.28, and 4.03 CFU, respectively. Interestingly, the survival of a pairing was associated with the microbial load of the constituent colonies, when considering only the colony of origin with the highest microbial load value ($P = 0.0009$) and when considering the cumulative microbial load level carried by both partners (i.e., the sum of the microbial load across the two colonies of origin) ($P = 0.0002$; Fig. 3a, b; Fig. S3). The better fit of logarithmic regressions in both analyses suggests that mortality as measured by hazard ratios only slightly increased

after a certain threshold of microbial load (Table S1). In outbred pairings that included an alate from colonies B or C, the failure of the pairings mostly resulted from the death of the alate from those colonies (Fig. 2c), consistent with their elevated levels of microbial loads (the daily number and origin of dead alates are provided in Fig. S2). In contrast, the opposite was found for outbred pairings including an alate from colonies A or D (low microbial loads), with the death of the partner originating from a different colony observed in most cases (Fig. 2c, d and S2). Finally, the relationship between the degree of relatedness of the partners and the hazard ratio of the colony pairing was not significant ($P = 0.666$), confirming the lack of effect of inbreeding on pairing survival during the first 14 days of colony founding (Fig. 3c). However, colonies C and E with the lowest number of surviving alates after 14 days (22 and 8, respectively; Fig. 2a) also exhibited high levels of relatedness (0.75 and 0.71, respectively), suggesting that these stock colonies were headed by inbred neotenics. In comparison, the degree of relatedness among members of the other colonies (i.e., A, B, D, and F) was close to 0.50, indicating they were probably headed by a monogamous pair of outbred primary reproductives (i.e., 0.48, 0.43, 0.52, and 0.54).

Metagenomic analyses revealed that bacterial communities were only slightly different between alates from different colonies (Fig. 4a); weighted UniFrac values did not separate individuals from different colonies, while unweighted distances only moderately did (Fig. 4a, b). Unweighted distances only consider the presence or absence of observed microbes, while weighted values also account for their abundance. Consequently, this results in similar levels of weighted bacterial differentiation observed within colonies and between different colonies ($P = 0.733$; Fig. S4), and a lower, but non-significant, level of unweighted differentiation within colonies than between colonies ($P = 0.381$). Fungal communities were also only moderately different between alates from different colonies (Fig. 4a). The level of differentiation between nestmate and non-nestmate alates was significantly lower for weighted values ($P = 0.045$), but similar for unweighted values ($P = 0.677$; Fig. 4a, b and S4). Overall, these results suggest that different colonies exhibit only slightly different bacterial and fungal communities. Consequently, the only unweighted fungal dissimilarity between partners is marginally associated with an increase in the Hazard ratio of their pairing ($P = 0.092$; Fig. 3g). However, the hazard ratio of a pairing was not associated with the weighted fungal similarity ($P = 0.261$), nor with the levels of either weighted or unweighted bacterial differences between partners (weighted: $P = 0.478$; unweighted: $P = 0.862$).

**Long-term survival of incipient colonies.** After a month, only 154 out of the 1421 alate pairings survived (10.84%), and only 85 (5.98%) survived until the fourth month (when the altricial larvae developed into workers able to provide care to both the parents and the next brood). Most of these colonies, 70 out of 85, survived until the end of the experiment (450 days, month 15): 33 were inbred and 37 outbred. Similar to the short-term survival, no significant difference was observed between the survival of inbred and outbred pairings over the course of the experiment ($P = 0.465$; Fig. 1b), while strong differences in survival were observed between specific pairings (Fig. S5). Notably, the hazard ratio of the different pairing combinations at 14 days was significantly correlated to that at 450 days ($P = 0.0009$; Fig. S6). This means that certain colony combinations were more likely to survive to both time points and that the development of brood and workers did not alter the ratio of surviving pairings after 14 days.

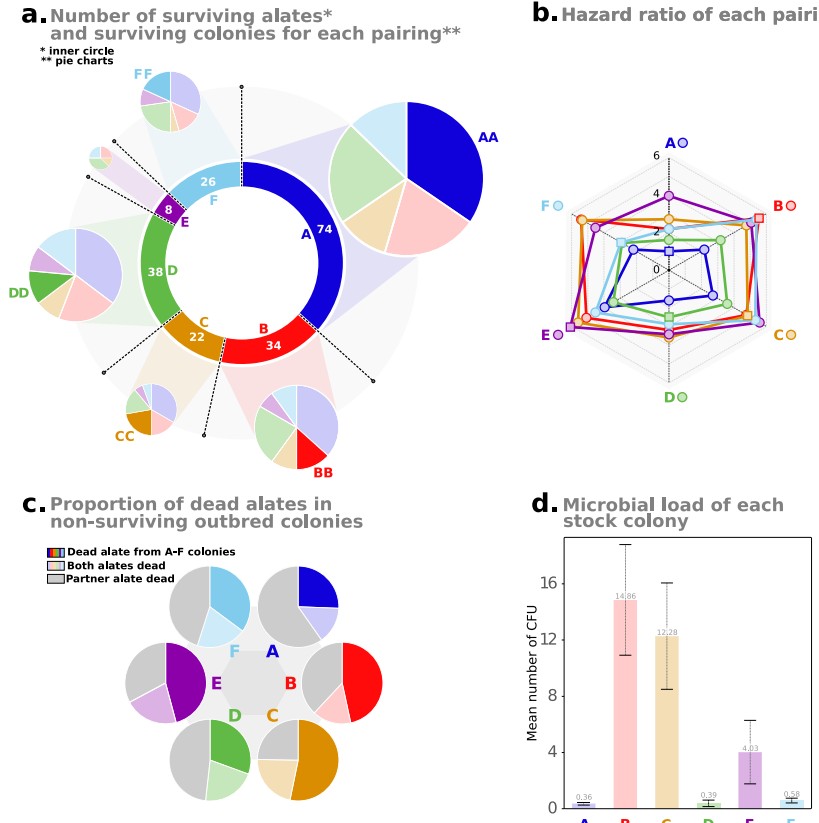

**Fig. 2 Survival and microbial load of alates according to their colony of origin. a** Colony of origin of the 202 surviving alates 14 days after colony establishment (inner circle). For each colony of origin, pie charts represent the distribution of surviving inbred and outbred pairings; outbred pairings are divided and light-colored according to the colony of origin of the partner, inbred pairings are represented by bright colors. **b** Radar plot represents the hazard ratio of each inbred and outbred pairings in the first 14 days after colony establishment. Parings with low hazard ratios (i.e., close to the center) are characterized by low mortality. Parings with at least one of the partners originating from stock colony A are colored in blue (B in red; C in orange; D in green; E in purple and F in light blue). Outbred pairings are marked with a circle, while outbred pairings are represented with a square. **c** Pie charts represent the proportion of dead alates in non-surviving outbred colonies for each stock colony. **d** Levels of microbial load of each stock colony, bars represent standard deviation.

**Productivity of inbred and outbred colonies**. Fifteen months after pairing, 68 of the 70 incipient colonies contained workers. The type of pairing significantly affected the number of workers present in colonies over time, with more workers produced in inbred colonies ($P < 0.001$; Fig. S7b); the mean number of workers was 25.06 ($\pm SD = 21.66$) in inbred colonies compared to 19.70 ($\pm SD = 21.16$) in outbred colonies (Fig. 5a). At the end of the experiment, 51 of the 70 colonies contained at least one soldier, with an average of 1.33 ($\pm SD = 1.17$) and 1.13 ($\pm SD = 0.93$) soldiers in inbred and outbred colonies, respectively (Fig. 5a). Similar to worker production, the type of pairing also significantly influenced the number of soldiers over time, with increased production in inbred colonies ($P < 0.001$; Fig. S7c).

**Survival and microbial load of inbred and outbred offspring**. In addition to estimating pairing survival, the microbial load and survival of their offspring were also monitored for 14 days following exposure to the entomopathogenic fungus *Metarhizium*. Inbred and outbred offspring differed in their survival when challenged with pathogens ($P = 0.001$), with inbred offspring exhibiting a higher mortality rate than those from outbred pairings (Fig. 5b). However, no significant difference was found between the microbial load of inbred and outbred offspring ($P = 0.401$; Fig. 5c), with the mean number of CFUs being 26.21 ($\pm SD = 19.14$) in inbred offspring and 30.93 ($\pm SD = 25.30$) in outbred offspring (Fig. S8).

## Discussion

Our study sheds light on the roles inbreeding and outbreeding play in the success of termite colonies over the course of their development. First, our results revealed comparable survival between inbred and outbred pairings during the first weeks of colony foundation, despite high survival differences between alates from different colonies. This suggests that inbreeding per se has no effect on survival at this stage of colony foundation; rather, the survival of the pairings is strongly influenced by the colony of origin of the constituent partners. The pairing with the highest survival was an inbred combination of alates from a low microbial-load colony, while the pairing with the lowest survival was also an inbred combination, but with alates from a high microbial-load colony (Supplementary Note 1). Our results show that the susceptibility of pairings increases with their cumulative and maximum levels of microbial load carried by the partners and only provides weak support for different colonies harboring distinct microbial communities; the survival of a pairing was only marginally associated with the fungal dissimilarity between partners. Together with the failure of pairings typically caused by the death of the partner with the highest microbial load, our results highlight the risk of unhealthy mate pairings, regardless of their level of relatedness. Yet, our results suggest that inbreeding takes its toll later when incipient colonies face pathogen pressure, as inbred offspring exhibit higher mortality toward pathogens. These findings suggest that although partner choice is initially influenced by the immediate advantage of

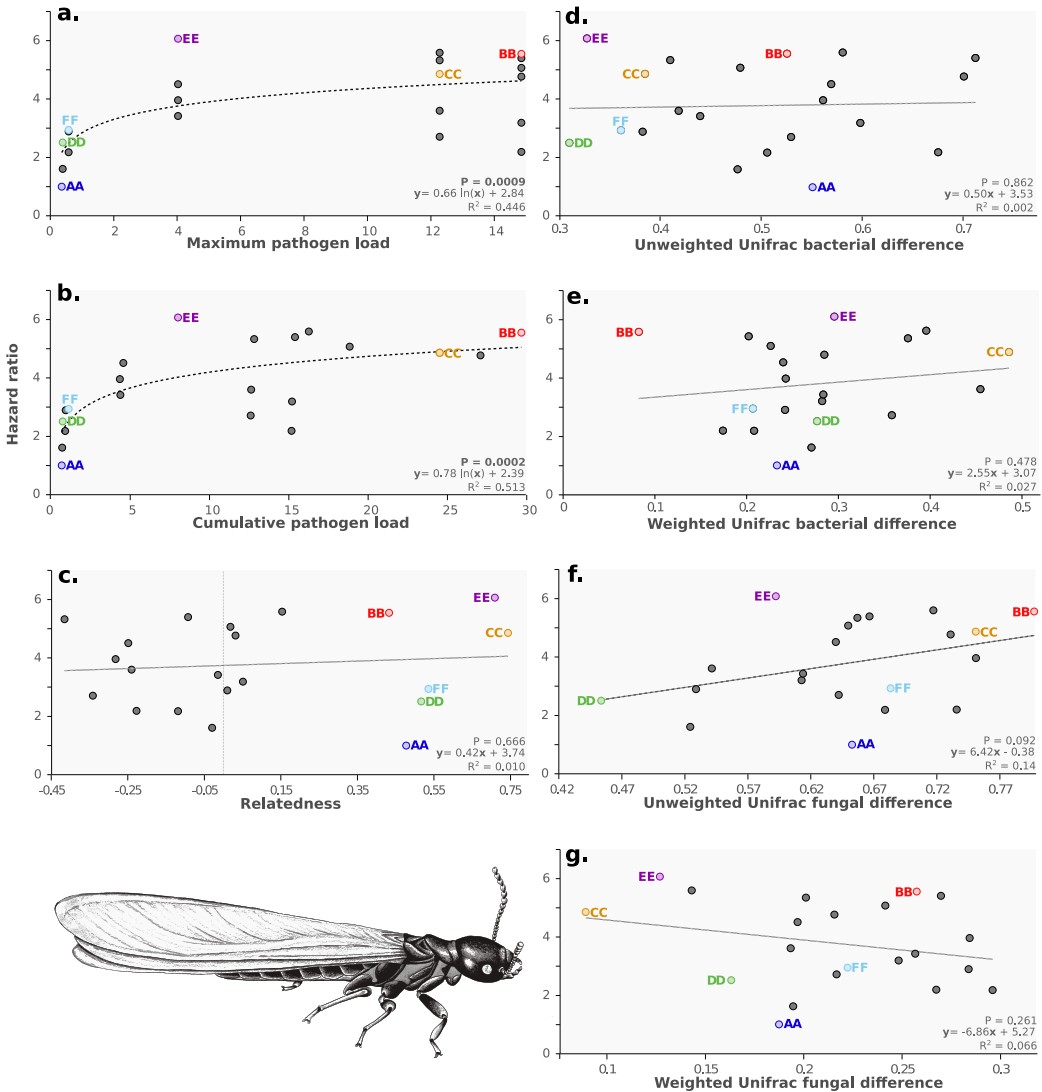

**Fig. 3 Factors influencing pairing survival.** Correlation between hazard ratio of a pairing and the maximum pathogen load (**a**), cumulative pathogen load (**b**), relatedness (**c**), unweighted Unifrac bacterial difference (**d**), weighted Unifrac bacterial difference (**e**), unweighted Unifrac fungal difference (**f**) and weighted Unifrac fungal difference (**g**). Trendlines represent logarithmic correlations for plots **a**, **b**, and denote linear correlations for all the other plots. In each plot, inbred pairings are colored according to their colony of origin.

a healthy partner rather than the long-term potential of more fit offspring, inbreeding depression during colony development may favor outbred colonies reaching maturity.

**Avoidance of a related or unhealthy partner**. Although an equal number of pairings for every pair of colonies was constructed experimentally, detection, and avoidance of partners who are either unhealthy (those with high microbial loads) or are nest-mates potentially occur during nuptial flights, discouraging random pairing in the field and minimizing the chance of pairing with a weak partner. We originally planned to test whether the choice of alates in this study relies on their level of relatedness, microbial similarity, and load (similar to refs. [35,36]). However, partner choice was inconsistent as alates either engaged in tri-tandem running or continuously changed partners (*pers. obs.*). To date, evidence of detection and avoidance of nestmate pairings are scarce and inconsistent in termites[12,37,38]. Inbreeding avoidance can occur through a split sex ratio between colonies, or differences between the sexes in their dispersal range or in their timing of emergence[39]. In termites, the low genetic similarity between neighboring colonies within populations[40,41] and the fact that alates fly away from their natal colonies[42,43] suggest that synchronous alate swarming is probably the predominant mechanism preventing inbreeding in many species (Note that alate dispersal is however often insufficient to maintain gene flow between populations[44,45]). Alates of most species do not seem to discriminate against nestmates, although this mechanism has been poorly studied[17]. Non-random matings despite long-range dispersal have been occasionally reported, with inbreeding avoidance in *R. chinensis*[36], but preference in *Coptotermes lacteus*[46] and *R. flavipes*[29]. Together with the large variation in the relatedness between partners occurring within and among species, and at different stages of the colony lifecycle (i.e., from colony foundation to mature colonies headed by neotenic reproductives in the case of subterranean termites and other lower termites)[17,47], our findings also support the conclusion that inbreeding avoidance is probably not a prime determinant of partner choice in termites during colony foundation[35].

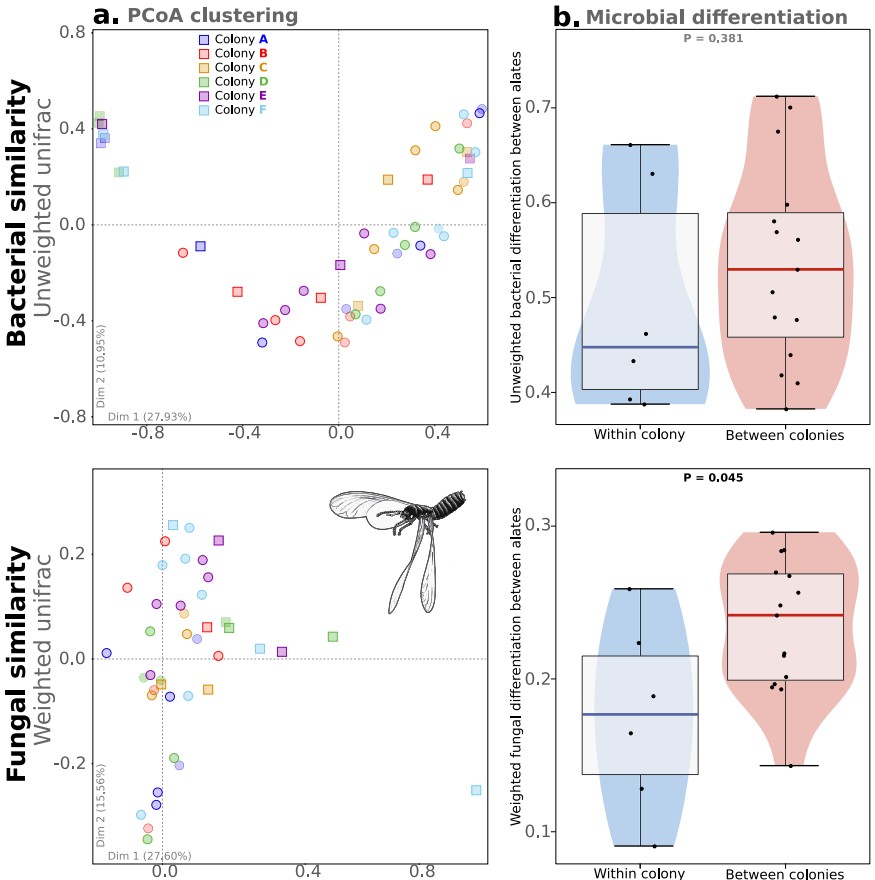

**Fig. 4 Microbial similarity within and among colonies. a** Principal Coordinate Analyses (PCoA) of individuals based on their unweighted Unifrac values for bacterial similarity and weighted Unifrac values for fungal similarity. Each individual is colored according to its colony of origin, alates are indicated with circles and workers with squares. **b** Violin plots of bacterial (unweighted Unifrac) and fungal differentiation (weighted Unifrac) among individuals within and between colonies. Box plots represent median and 1st and 3rd quartile; whiskers include 95% of all observations; dots indicate individual values. Results for weighted Unifrac bacterial similarity and unweighted Unifrac fungal similarity are provided in Supplementary Fig. S4.

Similarly, there is little evidence of detection and avoidance of unhealthy alates in termites, despite the fact that pathogen avoidance is commonly documented in workers[48–50]. In *R. chinensis*, alates paired less frequently with an injured partner[36], but females of *Z. angusticollis* showed no preference for healthy males rather than males infected with *Metarhizium*[51]. Our results revealed that the high risk of pairing with a sick partner represents most of the mortality observed during colony foundation, which suggests that pathogen recognition and avoidance should act as a strong selective force. This selection should not only be based on the detection of the external presence of spores, but on an overall evaluation of partner health, such as changes in behavior or cuticular hydrocarbons[52] (Supplementary Note 1). However, the influence of other potential selective pressures associated with nuptial flights (e.g., non-mating, predation and resource shortage) may instead lead partners to choose the first mate they encounter, regardless of their relatedness or health[53–55]. For example, most dispersing alates of the species *Odontotermes assmuthi* are lost through predation, which results in only 0.5% of flying alates surviving the nuptial flight[56]. In *Hodotermes mossambicus*, even after pairing and digging the first chamber, only about half of the de-alate pairs survive the first week[57]. Overall, these results highlight that choosiness is costly in termite, as extremely high predation pressure during colony foundation may act as a strong selective force to quickly find a mate and seek shelter[39,58,59].

**Offspring production**. Our results revealed a higher and faster production of workers and soldiers in inbred colonies. This result may be driven by the prevalence of inbred AA pairings and their weak microbial load. The higher productivity of inbred colonies (with low microbial load) may therefore stem from a tradeoff in resource investment between pathogen defense and offspring production[60]. In *Z. angusticollis*, pathogen pressure experienced by primary couples during colony foundation leads to a decrease in the likelihood of oviposition and the total number of eggs[19], and sibling pairs had higher survival than non-related couples when exposed to pathogens[61]. In *C. formosanus*, outbred pairings also suffered higher mortality than inbred pairings; but in this species, the decreased success of outbred pairings was offset by their increased productivity[62]. Importantly, most studies investigating differences in survival or productivity between inbred and outbred colonies have not used equal numbers of the various pairing combinations tested, nor taken into account the colony of origin (potentially testing for an interaction effect with the type of pairing). These studies may have failed to provide deeper insight into this process due to potentially strong differences between alates originating from different colonies and the lack of proper control to account for these differences. In our study, the equal pairing of every combination accounted for differences between colonies and resulted in similar survival between inbred and outbred pairings. However, a bias toward inbred or outbred colonies could be observed in the case of an association of alates from different colonies in different proportions (more inbred

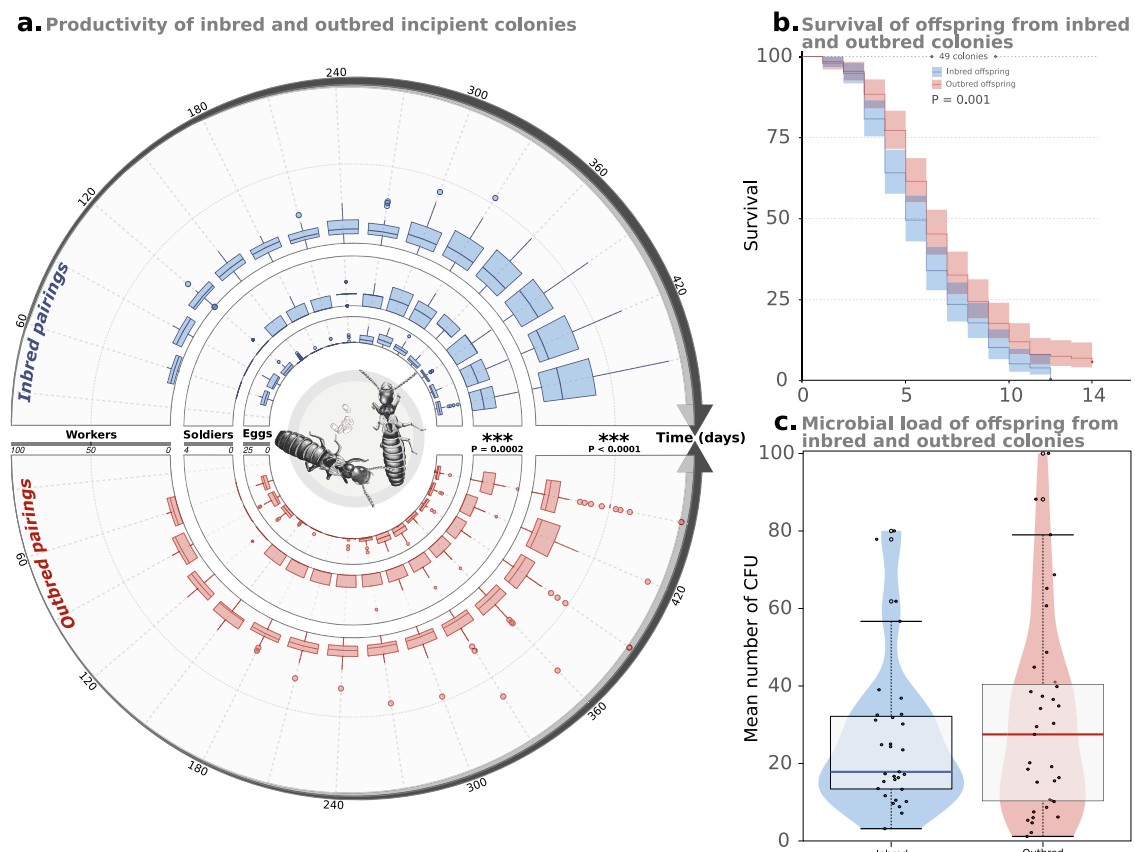

**Fig. 5 Productivity of inbred and outbred pairing, as well as survival and microbial load of their offspring. a** Graphical representation of the productivity of incipient colonies over the overall duration of the experiment (450 days, 15 months). Productivity is measured as the number of workers (outer circle), soldiers (middle circle), and eggs (inner circle) for each pairing. The productivity of inbred pairings is reported on the upper half-circle, while the productivity of outbred pairings is reported on the bottom half-circle. Box plots represent median and 1st and 3rd quartile; whiskers include 95% of all observations; individual dots indicate outlier values. *P* values indicate a significant effect of the type of pairing on the number of workers and soldiers in a colony over time, with increased production in inbred colonies (see also Supplementary Figure S6). **b** Kaplan–Meier survival distributions of offspring from inbred and outbred colonies when challenged toward entomopathogens. **c** Violin plot of microbial loads (mean number of CFU) of offspring from inbred and outbred colonies. Box plots represent median and 1st and 3rd quartile; whiskers include 95% of all observations; dots indicate individual values.

pairings from the healthy colony A and less from the susceptible colony E would have resulted in better survival of inbred pairings compared to outbred pairings).

**Offspring survival**. Our results show that incipient colonies may suffer from inbreeding when facing pathogen pressure, although cuticular microbial loads did not differ between inbred and outbred offspring. In contrast, higher microbial loads were observed in inbred colonies of *Z. angusticollis*, potentially resulting from reduced grooming or less-effective antimicrobials[28]. Notably, the higher mortality of inbred offspring in our study contrasts with the absence of an inbreeding effect on the survival of the pairings (parents) over the 15-month study period. This difference may potentially stem from the high pathogen load experimentally used to assess offspring mortality. Similarly, the absence of an effect of inbreeding on the survival of the pairings may also reflect the low and homogeneous pathogen pressure that pairings experienced under lab conditions during colony founding. Our findings however suggest that, under a more diverse pathogen pressure naturally occurring in the field, the reduced survival of inbred offspring in incipient colonies progressively decreases the proportion of inbred pairings over time. Our results on incipient colonies also contrast with those uncovered in mature field colonies of the same species, showing a weak influence of genetic diversity toward entomopathogens[30,31]. First, this difference may stem from a

greater reduction in heterozygosity in the present study compared to those in mature colonies, where heterozygosity was only moderately reduced by neotenic reproduction[24,30]. Similarly, offspring in the present study were probably younger and thus more susceptible to pathogen exposure[23]; they were also reared under lab conditions and did not face the same pathogen exposure as workers collected from the field, therefore removing the possibility that immune priming may potentially mask differences between inbred and outbred groups[63,64]. Despite these differences, the better survival of particular pairings also supports the suggestion that the influence of a specific genetic background may be greater than the overall genetic diversity on colony survival[30,31]. Together with previous findings, our results reveal that inbreeding is a negligible factor in the survival of both founding couples and mature colonies; but may have an important role in incipient colonies under conditions of high pathogen load. These findings indicate that higher inbreeding depression during colony development, where incipient colonies may be more vulnerable, could increase the proportion of mature colonies headed by outbred reproductives[29] (illustrated in Fig. 6).

**Inbreeding is only a risk for small incipient colonies.** Inbreeding acts differently upon colonies depending on their stage of development, and may therefore not play an important role in partner choice. Inbreeding depression only occurs in small

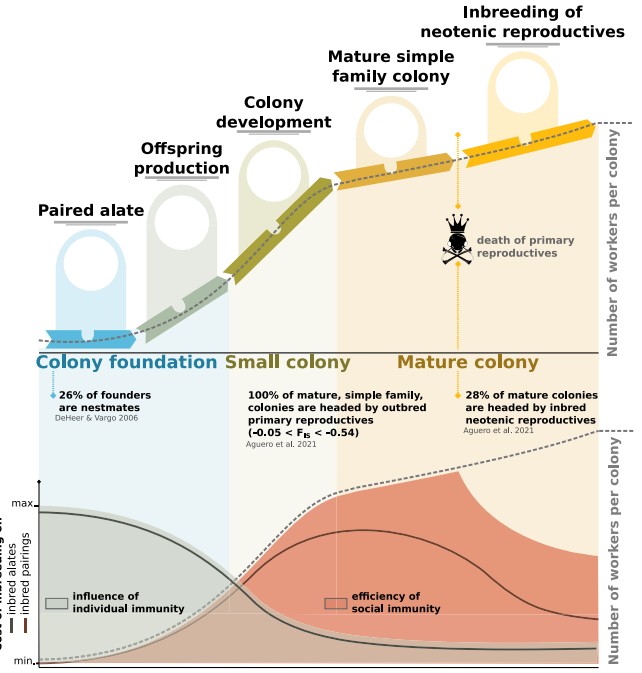

**Fig. 6 Schematic illustration of the cost of inbreeding termite colonies face over the different stages of their lifespan.** The dotted lines represent colony size (i.e., the number of workers per colony). The red line represents the cost of inbreeding depression on inbred pairings. Inbreeding depression is low during colony foundation and offspring production, but is higher during colony development, when small colonies face pathogen pressure (this study[29]). The red area represents the efficiency of social immunity, which increases with colony size until it is expected to slightly decrease due to inbreeding from neotenic reproduction. The gray area represents the influence of individual immunity, which is high in founding couples and in small incipient colonies. The cost of inbreeding in inbred alates (gray line) is high before incipient colonies become large enough to benefit from social immunity. The high efficiency of social immunity in large mature colonies releases inbreeding depression, allowing the development of inbred neotenic reproductives without suffering costs associated with pathogen pressure[30].

colonies. In our study founding, couples experienced drastic mortality in the first weeks, even though the risks associated with nuptial flights mentioned above were limited under laboratory conditions. The presence of strong selection against inbreeding during pairing is also discredited by the common occurrence of inbreeding through neotenic reproduction observed in mature colonies. Remarkably, while inbreeding is prevented in vertebrate social species via parental inhibition of sexual activity by the parent of the opposing sex, the opposite is found in termites. The removal of reproductives triggers the development of same-sex reproductives and sometimes fosters the development of opposite-sex reproductives, therefore promoting inbreeding to maintain the life of the colony[65]. The frequent inbreeding in mature termite colonies suggests a reduced level of inbreeding depression. Reduced inbreeding depression may in fact be a consequence of frequent mating between neotenics, as the occurrence of low levels of inbreeding within populations is expected to result in a purge of deleterious alleles over time[66–69]. Similarly, the reduced inbreeding depression in termites may result from their specific sex-determination system based on heterochromosomes[70]. A substantial part of the genome in some termites (sometimes over 50%; possibly four to eight out of the 42(2n) chromosomes in *Reticulitermes* males[71,72]) is sex-linked, whereby the Y chromosome and some autosomes segregate together as a single linkage group. This feature leads to the formation of chains of chromosomes inherited together during meiosis. Under male heterogamy (XY = male), autosomes linked to the Y chromosome never become homozygous by descent in the absence of crossing-over, allowing heterozygosity to be conserved across the large sex-linked portion of their genome[73,74]. This specific sex-determination system, therefore, helps termite species to reduce genetic costs associated with inbreeding in males (usually 50% of the worker force). Finally, in a few termite species, inbreeding is largely avoided through the production of neotenic queens via parthenogenesis, and their interbreeding with the original primary king[75–77].

Neotenic inbreeding may be tolerated in populous colonies, when social immunity becomes more important than individual immunity in managing pathogen pressure[21,78–81]. Social immunity in termites strongly relies on allogrooming, cannibalism, burial behavior, and self-exclusion of infected individuals[82,83]. Although these behaviors may be adequate for mature colonies, they may be costly in incipient colonies, and cannot be applied to reproductive individuals. These behaviors may therefore be more prevalent and efficient in large groups[24], accounting for the higher influence of individual immunity (related to individual genetic diversity as determined by inbreeding) in small incipient colonies. Likewise, the primary couple also lacks the benefits of social immunity in the initial stages of colony foundation, suggesting that the individual immunity of the founders also plays an important role. In our study, alates from inbred stock colonies (C and E; probably headed by neotenics) suffered high mortality after 14 days, in comparison to alates originating from stock colonies headed by outbred primary reproductives. Hence, although social immunity may allow neotenic inbreeding in populous colonies, those colonies may suffer from producing inbred alates with reduced individual immunity that will not survive long enough to benefit from social immunity that occurs when workers are produced. Interestingly, individual immunity is negatively correlated with colony-level immune behaviors in an ant, suggesting a trade-off between individual and social immunity in regulating overall parasite protection in this species[84]. Similarly, the development of social immunity in shaping disease resistance in termites (also in social Hymenoptera[85]) is hypothesized to occur at the expense of individual immunity, as the evolution of sociality is associated with a reduction in their immune gene repertoire[86–88] (but see refs. [89,90]).

Although inbreeding avoidance is an appealing concept in evolutionary biology, evidence is scarce for its widespread occurrence[91], with mate choice encompassing the entire spectrum from inbreeding preference to tolerance to avoidance[92]. This variability is observed both within and between species, and is related to the strength of inbreeding depression[93]. Individuals would not be selected to avoid mating with a related partner if the chance and costs of inbreeding are low and if the costs associated with nestmate discrimination are high[94]. For example, our findings may not apply to most social Hymenoptera, due to the extra cost of inbreeding resulting from their haplodiploid sex determination, in which a single founding queen cannot afford the burden of producing up to 50% workless and sterile diploid males[95,96]. In contrast, the common occurrence of inbreeding among neotenics in mature termite colonies suggests a lower level of inbreeding depression. Overall, our findings emphasize the varied and changing costs of outbreeding and inbreeding and how these play out over the lifespan of termite colonies. Investigating this variation and its costs will surely provide insights into the evolutionary mechanisms driving inbreeding avoidance and preference in social insects.

## Methods

**Termite collection and alate pairing**. Six stock colonies (colonies A to F) of *Reticulitermes flavipes* were collected in Bryan, TX, USA in March 2020, a week before the swarming flight would have naturally occurred. Colonies were extracted from their wooden logs and transferred into 20 cm plastic boxes. One worker per colony was sequenced at the mitochondrial 16 S gene to confirm identity of the species, following methods from Aguero et al.[31]. Within a week after collection, male and female alates were sexed for each colony and isolated with a group of nestmate workers. They were then paired in 3-cm petri dishes with sawdust and wood pieces[97]. The incipient colonies were kept in high humidity chambers. Only dark-pigmented alates were used to ensure they were physiologically and motivationally ready to mate.

To investigate the short-term effect of inbreeding on founding success, we set up 40 inbred pairings for each colony (only 31 for colony D due to a lack of available alates). We also prepared 40 outbred pairings for every combination of colonies, with an equal number of each sex per colony of origin (20 queensA x kingB and 20 queensB x kingA); resulting in 231 inbred and 600 outbred incipient colonies. In addition, we estimated the long-term effect of outbreeding on incipient colony survival and productivity, as well as on pathogen resistance and microbial load of their offspring. To ensure robust sample sizes, we anticipated high mortality during colony foundation and established an additional 290 inbred and 300 outbred pairings (100 inbred pairings for three colonies with enough alates available: colonies A, B & F, only 90 inbred pairings for colony F; and 100 outbred pairings for all combinations of those colonies). Overall, we set up 1421 incipient colonies (521 inbred and 900 outbred), all of which were established on the same day.

**Relatedness between colonies of origin**. For each stock colony, DNA from 10 workers was extracted using a modified Gentra PureGene protocol and genotyped at nine microsatellite loci[30]. Amplifications were carried out in a volume of 10 μl including 1 U of HS DNA polymerase, 2 μl of 5× buffer (MyTaq™, Bioline), 0.08 μl of each primer, and 1.25 μl of DNA template. PCR was performed using thermocycler T100 (Bio-Rad). Alleles were sized against a LIZ500 standard on an ABI 3500 genetic analyzer (Applied Biosystems) and called using Geneious v.9.1[98].

Relatedness coefficients (r) and their variances were estimated among nestmates and between workers from each pair of colonies using the Queller and Goodnight[99] algorithm implemented in the program COANCESTRY v.1.0[100]. A principal component analysis was performed on the microsatellite markers using the *adegenet* package[101] in R Development Core Team to visualize and confirm genetic differentiation between sampled colonies (Fig. S1).

**Microbial-load estimation**. For each stock colony, microbial loads were estimated from the number of CFUs cultured from individual cuticular washes of 12 alates (6 females and 6 males) and 6 workers per colony. Each alate was washed in a sterile 1.5 ml tube with 300 μl of a 0.1% Tween 80 solution, gently vortexed and centrifuged at $300 \times g$ at 4 °C for 20 minutes[102]. For each sample, three 20 μl replicates of the supernatant were plated on potato dextrose agar, while 20 μl of the Tween 80 solution was used as a control. Plates were inverted and incubated at 37 °C for three days. The number of CFUs at least 1 mm in diameter was counted for each plate and averaged between triplicates. Microbial loads were quantified the same day as the alates were paired. Microbial loads were compared between colonies using a Mann–Whitney U-test. For each pairing combination, cumulative microbial load describes the sum of the microbial load across the two colonies of origin, while maximum microbial load only considers the colony of origin with the highest value.

**Microbial diversity identification**. Bacterial and fungal communities were identified for each colony by sequencing cuticular washes of three female alates, three male alates, and three workers per colony ($N = 54$). Individuals were collected using sterile tools and washed in 300 μL of 0.1% Tween 80 solution. After 15 minutes of gentle rotation, the solution was removed for DNA extraction using a Phenol/Chloroform protocol. For the bacterial community, the v4 hypervariable region of 16 S was amplified using the bacterial primers 515 f and 806r[103]. For the fungal community, ITS was amplified using the primers CS1-ITS3 and CS2-ITS4 with Fluidigm CS1 and CS2 universal oligomers added to their 5′- end[104]. PCR protocols are provided in Supplementary Methods[105]. Pooled amplicons were loaded onto an Illumina MiSeq Standard v2 flow cell and sequenced in a $2 \times 250$bp paired-end format using a MiSeq.v2.500 cycles reagent cartridge. Base calling was performed by Illumina Real Time Analysis v1.18.54 and output was demultiplexed and converted to FastQ format with Illumina Bcl2fastq v2.19.1. All analyses were performed using *QIIME 2*[106]. Paired-end reads were filtered for quality control and combined using the *DADA2* pipeline[107]. 16 S and ITS sequences were joined at 250 bp and identified as amplicon sequence variants. Samples with low coverage (<10,000 reads) were removed from further analyses; all samples were conserved for bacterial analyses, but 13 samples were discarded from fungal analyses. To estimate microbial difference within and between colonies, weighted and unweighted UniFrac distances between each individual were visualized using a principal coordinates analysis (PCoA)[108]. Unweighted distances only consider the presence or absence of observed microbes, while weighted values also account for

their abundance. Euclidean distances between pairs of individuals on the two PCs of the PCoA were used to build pairwise distance matrices and to compare differentiation among individuals within and between colonies using a Mann–Whitney U test.

**Short-term cost of outbreeding**. The survival of the 231 inbred and 600 outbred colonies was assessed every two days for 14 days after pairing. The additional 590 colonies were not used for this experiment because they were only monitored once a month (see below). For each unsuccessful colony (i.e., at least one reproductive died), the sex of the dead alate was assessed to determine its colony of origin. Survival distributions were compared between inbred and outbred pairings and between pairings using the *Coxph*-proportional Hazards model implemented in the *survival* package[109] in R. This model was also used to calculate hazard ratios for each colony pairing. Linear and logarithmic regressions were performed to determine the relationships between the hazard ratio of each pairing and the effect of the relatedness between partners (microsatellite analysis), cumulative microbial load, maximum microbial load, as well as fungal and bacterial similarities.

**Long-term cost of inbreeding**

*Survival and productivity of incipient colonies.* The survival of the 1421 alate pairings (521 inbred and 900 outbred) was assessed every month for 15 months. Survival distributions were compared between pairs of colonies of origin, as well as between inbred and outbred pairings using the *Coxph* model. The productivity of all surviving colonies was assessed monthly by counting the number of eggs, workers and soldiers. The difference in productivity between inbred and outbred pairings was determined using two generalized linear models implemented in the *lme4* package[110] in R. The models tested the relationship between the numbers of workers and soldiers present in colonies as a function the type of pairing (inbred or outbred), with time tested as a covariable. The number of eggs present in a colony was not used because of its bimodal distribution (absence during winter) and non-cumulative nature (eggs "disappear" once they hatch). Linear regression was performed to determine the relationship between the hazard ratios at 14 days and at 15 months after pairing of each combination of colonies.

*Survival and microbial load of the offspring produced.* After 15 months, just 70 out of the 1421 incipient colonies survived, of which only 49 produced 10 or more workers. For each of the 49 colonies (24 inbred and 25 outbred colonies), a group of eight workers were isolated in 30 mm petri dishes lined with filter paper (Whatman Grade 5, porosity 2.5 μm). Groups were challenged with a pathogen solution containing three strains of *Metarhizium* fungus in equal proportions at the concentration of $1 \times 10^7$ conidia/ml in 0.1% Tween 80 (ITS sequences match accession numbers KU187187.1, MT374162.1 and LT220706.1, for *M. anisoplae*, *M. brunneum,* and *M. guizhouense*, respectively). Offspring survival was monitored for 14 days following exposure by moistening the filter paper with 300 μL of the fungal solution[30]. Difference in survival between inbred and outbred offspring was determined using the *Coxph* model. In addition, 66 of the 70 incipient colonies had at least two workers (31 inbred and 35 outbred colonies), for which two workers (with three replicates each) were used to determine the microbial load of the offspring. Microbial loads were measured as described above, except that cuticular washes of workers were extracted in 100 μL of a 0.1% Tween 80 solution.

**Reporting summary**. Further information on research design is available in the Nature Research Reporting Summary linked to this article.

## Data availability

The data reported in this study have been deposited in the Open Science Framework database, https://osf.io. https://doi.org/10.17605/OSF.IO/CA4HD.

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

## Acknowledgements

We are grateful to C. Aguero and M.N. Moran for their help in incipient colony establishment, as well as with the microbial loads experiments. We thank M. Bulmer for providing the *Metarhizium* fungal strains. This work was supported by the Urban Entomology Endowment at Texas A&M University.

## Author contributions

P.A.E. and E.L.V. designed the study. P.A.E. collected the samples, performed the experiments and analyzed the data. P.A.E. wrote the paper with the contributions of E.L.V.

## Competing interests

The authors declare no competing interests.
