## [Peer Review File · Communications Biology]

Reviewers' comments:

Reviewer #1 (Remarks to the Author):

In this manuscript, authors investigate colony foundation success of inbred versus outbred pairs in the termite *Reticulitermes flavipes*. They show that both are susceptible to costs, but at distinct stages of the colony lifespan. In the short-term, colony foundation is more susceptible to overall partners' health, while its survival in the long-term will be function of their offspring's genetic diversity. Interestingly, inbred colonies seem to perform better (they produce more offspring) in the absence of pathogen pressure. The paper is well-written and of high quality, impact and relevance to the field. I have only a few comments.

Context on genetic peculiarities of termites are missing, but are of importance for their consequences on heterozygosity. It is only at L349 that any mention to how sex determination is achieved in social insects (but for Hymenoptera). It should be stated early on that termites are diplo-diploid, with their sex determination seemingly based on heterochromosomes. Part of their genome (sometimes over 50%) is sex-linked, males being translocation heterozygotes. This happens through the formation of multivalent chains or rings of chromosomes during meiosis, only in males. Typically, four out of the 42 chromosomes are combined into a multivalent chain in *Reticulitermes* males. It follows that a differential susceptibility/mortality might occur in the offspring, both sexes being represented in the work force. Additionally, inbreeding through neotenic reproduction (L322) is also avoided in several species through the production of neotenic queens via parthenogenesis.

Authors show that of 12% of pairs (101 on 831) survived after two weeks. I find that some figures for comparison are generally missing from the main text and would greatly benefit the reader. As such, around 50% of settled pairs survive the first week in *Hodotermes mossambicus*, (Darlington et al., 1977. *Insect. Soc.* 24: 353–358). This represents very little, most dispersing alates being lost at an earlier stage mostly through predation: only 0.5% of flying alates would survive the nuptial flight in *Odontotermes assmuthi* (Basalingappa, 1970. *Indian Zool.* 1: 45–50). Although not directly comparable (field observations versus lab ones), I believe these would nicely complement the global picture on colony foundation that the authors depict in their discussion. Another result is the survival of colonies after 15 months, with only 5% (70 of the 1421) of colonies remain. Of these, 49 produced enough workers to be considered in further analyses (L460-461): 24 were inbred and 25 outbred. What about the remaining 21? Please clearly state the breakout for the 70 colonies in section starting at L179.

Minor comments

L53-54: "...scent of related males...in their female relatives.". "related" can be dropped.

L114: Pairings.

L133: susceptibility of/to what?

L154: "..., level of unweighted...".

L160-161: Please correct "...marginally-significantly associated..." to "...marginally associated...".

L207: "... a high microbial-load.." but not the highest one. Please specify.

Figure 4: please add on the Figure the colour code of colonies.

Figure 4a: something seems written in upper left corner?

Reviewer #2 (Remarks to the Author):

Review of Short and long-term costs of inbreeding in the lifelong-partnership in a termite, by Eyer and Vargo

This is an interesting, well designed and thorough study of the effects of inbreeding in a species of termites. Somewhat unexpectedly, foundational success of colonies was not influenced by relatedness between partners, but by their microbial loads. Inbreeding depression does take its toll later in colony development, as inbred workers are more susceptible to entomopathogens, although colony survival was found to be positively correlated with the level of inbreeding (see comment 3). I think the analyses are generally well done, and the paper is well written. However, some of the results are unexpected, which makes it hard to explain for the authors, and to follow

the logic for the reader. I also feel that the authors are strongly guided by an expectation of inbreeding depression, which perhaps biases some of their interpretations. I think this is unnecessary, and in fact, the regular occurrence of inbreeding by replacement reproduction may give an alternative expectation, which may be discussed (see comment 8). I have some general comments and some suggestions for improvement:

1. The conclusions that inbreeding takes its toll later in life is an inference based on the finding that inbred workers have higher sensitivity to pathogens. This result is in conflict with the direct observation that the survival of colonies was not negatively affected by inbreeding, in fact on the contrary. Please specify this conclusion, both in the abstract and elsewhere in the manuscript.
2. Lines 46-54: Different ways to reduce inbreeding are mentioned. I thought a very obvious one was lacking, the highly synchronised swarming seen in many species.
3. In the results section it is not specified what is meant with inbred and outbred colonies. As far as I can see, inbred colonies are established from sib-alate pairings, and outbred from pairings between alates from different colonies. Please add this information and those results at the start of the results section.
4. I also see that relatedness between different source colonies has been estimated using microsatellites, but nowhere is this result being discussed in the result section. Please specify.
5. Line 190: Here the survival of inbred and outbred workers when challenged with pathogens is described. It took me a while to understand the difference with the first results on initial colony survival. It would help if this experiment is introduced with the details of the entomopathogens used and to describe that individual survival was measured.
6. Line 220: The title of this section of the discussion is "Avoidance of inbred or unhealthy partner" I think what is meant here is "Avoidance of related or unhealthy partner". (Inbred partner could be a separate question, not addressed in this study)
7. I found figure 6 not very clear. Especially the lower part: it contains an X and a single Y axis, but three different dependent variables. Perhaps an additional axis on colony size can be specified on the righthand side of the figure, and then the coloured part can be explained.
8. Lines 350-351: "In contrast, the common occurrence of inbreeding among neotenics in mature termite colonies suggests a lower level of inbreeding depression." While this is true, the authors could perhaps add that the low level of inbreeding depression may in fact be a consequence of frequent mating between neotenics, as this will purge deleterious alleles. So in other words, frequent inbreeding makes them resistant to inbreeding.

Reviewer #3 (Remarks to the Author):

The authors investigated the cost of inbreeding in the termite *Reticulitermes flavipes*, paying special attention to the importance of pathogens. Starting with a large number of termite couples set up in the lab either under inbred or no-inbred conditions, their results showed that inbreeding does not seem to have fundamental negative effects on the colonies' success. Inbred colonies grew faster during early colony development but could cope less well during experimental infections with pathogenic *Metarhizium* fungi.

The experiments are comprehensive and well done. The authors add with their study new data to a controversial topic. Thus, the study is of relevance for people from the field but also beyond. The latter aspect can be worked out more in a revision. I have some points that can and should be addressed in a revision.

Major comments

Some revision of the text is needed. Some sentences are unclear or illogical (see below). Some precision of terms and concepts are required: Outbreeding, inbreeding depression.. Furthermore, some sentences should be more carefully phrased as they are based on results from a single/few species, neglecting the diversity in termites; see below.

In addition, I have some statistical and methodological comments that should be considered.

The title should be adapted; given the lifespan of colonies the experiments were not long-term.

Although the termite drawings are nice and might work in an oral presentation, they should be

removed in the manuscript. They are too prominent and distract from the main content of the figures.

Some figures are unnecessarily complicated. Please, simplify (see below).

Details and Other comments

Methods & Stat.

Relatedness estimates: Please provide confidence intervals for relatedness estimates. Figure S1 requires better explanation. What do you mean with relatedness coefficients were weighted equally? What was the background allele frequency against which relatedness was estimated?

I.431: Please correct for multiple testing

I. 454-456: Time should be a covariate not a factor, and 'type of pairing' a fixed factor as you manipulated it purposefully (even though you arbitrarily assigned pairs, I guess)

Others

I.28-29: Unclear sentence. Seem illogical, how can a faster growth (something positive) reveal a trade-off.

I.31-32: unclear sentence. Why 'although'? Change conclusion: I think it is also not correct to say that outbreeding is generally favoured. (i) this only applies with pathogen infection, and (ii) it would mean that inbreeding is disfavoured. Finally, the colonies were not tested up to the stage of maturity (i.e., when the first sexual offspring are produced).

I. 44- 54: This section – as well as the discussion – overlooks purging selecting: the removal of slightly deleterious alleles over time when inbreeding commonly occurs. Which is probably the case, given that termites commonly produce neotenic. A fact that is often overlooked in the termite literature.

I.57-58: The opportunity for extra-pair fertilizations largely depends on the taxonomic group/life type in termites: in wood-dwelling dampwood and drywood termites it is very common see e.g. termite review: Korb & Thorne 2017, and work on *Z. nevadensis* by Thorne group (e.g. Thorne et al. 2003), and on *Cryptotermes secundus* by Korb group (Schneider & Korb 2007, Korb & Roux 2021)

I. 65: This should be written more carefully because this conclusion is just based on Zootermopsis, as a dampwood termite it is exposed to exceptional high pathogen loads, which differs e.g. compared to drywoods.

I.69-71: well both partners are primarily also required for mating as only a minority of species can re-produce parthenogenetically.

I. 87 and ff.: outbred pairings must be defined. Often 'outbreeding' means less closely related than expected by chance. This is not the case here; here it is rather non-inbred.

I.98: and in the title: 15 months is not really longterm given the longevity of queens/kings /the colony. Please change throughout

The introduction is missing the important part that choosiness is costly, especially in termite which have experience extremely high predation pressure during colony foundation. This should also be more stressed in the discussion. In the field alates/dealates are just eaten by predators; every extra minute outside means a very high risk to be eaten. This strongly selects against choosiness in termite which is also often neglected in such discussion.

Results: As the methods are presented after the Results, more information needs to be given in the results so that a reader can understand: e.g. (i) the name of the statistical tests, incl. what was specially tested: random factor, fixed factor..; (ii) explain what Unifrac values are and the difference between weighted and unweighted, (iii) explain highest/cumulative microbial load..

I.114: typo: pairings

Figure 2. This figure is unnecessarily complicated and it is hard to see the important results. (a) and (c) can, e.g. be just presented as bar plots. Pie charts not very useful, exchange them for staged bars. (e.g. for c. colonies along the x-axis, pie charts as staged bar. The radar plot is incomprehensible and not explained.

I am wondering about the relatedness values. Figure 3 implies that colony C and colony E are inbred colonies as r is around 0.75, while colony B below 0.5 (fused colony?). Is this correct? This should be discussed.

I.168 add %

I.174: Which test was applied?

I.215: Here and elsewhere: Choosiness was not investigated. The experiments cannot tell whether the termites are in nature choosy. Please, adjust.

I.281: unclear, what does 'restore' mean? Note, at 15 months the colonies are not mature.

I.222: susceptible is the wrong term

I.220 and ff.: The study cannot say whether termites are choosy in nature. The importance of predation which is a major driver in nature should be pointed out more in this section.

I.231: there is important literature missing. Actually these indirect mechanisms do not seem to be rare. See major important review by Nutting 1969. Females very generally disperse before males.

I.232-234: please also read Nutting 1969 for this. What is genetic differentiation between colonies? Here more real population genetic studies should be cited that measured genetic differentiation e.g. work by Kaib & Brandl, Goodisman, Schmidt & Korb.

I.239: what is meant with different stages of a colony lifecycle

I.248: predation acts even as a stronger selective force which can be seen when dispersal is studied in nature (Nutting 1969, Korb & Schneider 2007, Korb & Salewski 2000).

I.252-253: see above, links to relevant termite literature should be provided.

I.258: unclear sentence

I.265 and ff.: I agree completely that the colony of origin is an important factor that was not considered in other studies. Yet, equal numbers of colonies in set up are less important. This is taken into account with the statistical analyses, when you include colony of origin as a factor. One could /should even include the interaction between 'Colony of origin' and 'inbred: yes, no' as factors. This will reveal these interactions.

I.279-281: If you did not find differences in microbial loads between inbred and non-inbred colonies, than you cannot explain this with results that in another study these differences were found.

I.288: reference format.

I.293-298: important to add 'under conditions of high pathogen load'

I.324 and ff: unclear. Also in vertebrates, individuals prevent same sex individuals from reproducing breeding.. same in social insects. The removal of reproductives trigger development of same sex reproductives and sometimes foster the development opposite sex reproductives (but only if there is none.. Add References

I.303 and ff: this section should include considerations of purging selection (major disadvantages of inbreeding disappear via purging selection) and predation.

I.338-339: the reference numbers are assigned correctly. Furthermore, the statement is not correct as currently evidence seem equal strong for both hypotheses.

I.350: This depends on the exact mode of sex determination in Hymenoptera. As is now known it can be zygoty at specific loci that matter.

Judith Korb

References

- Korb J, Salewski V (2000) Predation on swarming termites by birds. *African Journal of Ecology* 38:173-174
- Korb J, Schneider K (2007) Does kin structure explain the occurrence of workers in the lower termite *Cryptotermes secundus*? *Evolutionary Ecology* 21:817-828
- Korb J, Roux EA (2012) Why join a neighbour: Fitness consequences of colony fusions in termites. *Journal of Evolutionary Biology* 25:2161-2170.
- Korb J, Thorne B (2017) Sociality in termites. In: *Comparative Social Evolution* (eds. DR Rubenstein, P Abbot) Cambridge University Press, pp. 124-152
- Thorne BL, Breisch NL, Muscedere ML (2003) Evolution of eusociality and the soldier caste in termites: influence of intraspecific competition and accelerated inheritance. *Proc Natl Acad Sci USA* 100:12808-12813

Reviewer #1 (Remarks to the Author):

In this manuscript, authors investigate colony foundation success of inbred versus outbred pairs in the termite *Reticulitermes flavipes*. They show that both are susceptible to costs, but at distinct stages of the colony lifespan. In the short-term, colony foundation is more susceptible to overall partners' health, while its survival in the long-term will be function of their offspring's genetic diversity. Interestingly, inbred colonies seem to perform better (they produce more offspring) in the absence of pathogen pressure. The paper is well-written and of high quality, impact and relevance to the field. I have only a few comments.

Context on genetic peculiarities of termites are missing, but are of importance for their consequences on heterozygosity. It is only at L349 that any mention to how sex determination is achieved in social insects (but for Hymenoptera). It should be stated early on that termites are diplo-diploid, with their sex determination seemingly based on heterochromosomes. Part of their genome (sometimes over 50%) is sex-linked, males being translocation heterozygotes. This happens through the formation of multivalent chains or rings of chromosomes during meiosis, only in males. Typically, four out of the 42 chromosomes are combined into a multivalent chain in *Reticulitermes* males. It follows that a differential susceptibility/mortality might occur in the offspring, both sexes being represented in the work force.

RESPONSE: We agree with this comment and we now provide information on the specific sex determination system of termites that may help reduce the cost of inbreeding in males. We now indicate in the Introduction that '*termites are diplo-diploid eusocial insects [...]*' (see *Introduction, Lines 58*). In the Discussion, we now explain that '*the reduced inbreeding depression in termites may result from their specific sex determination system based on heterochromosomes. A substantial part of the genome in some termites (sometimes over 50%; possibly four to eight out of the 42(2n) chromosomes in Reticulitermes males) is sex-linked, whereby the Y chromosome and some autosomes segregate together as a single linkage group. This feature leads to the formation of chains of chromosomes inherited together during meiosis. Under male heterogamy (male = XY), autosomes linked to the Y chromosome never become homozygous by descent in the absence of crossing-over, allowing heterozygosity to be conserved across the large sex-linked portion of their genome. This specific sex determination system therefore helps termite species to reduce genetic costs associated with inbreeding in males (usually 50% of the worker force)*' (see *Discussion, Lines 378-387*).

In addition, we now provide the associated references:

Yashiro T et al. 2021. *Enhanced heterozygosity from male meiotic chromosome chains is superseded by hybrid female asexuality in termites, PNAS.*

Charlesworth & Wall. 1999. *Inbreeding, heterozygote advantage and the evolution of neo-X and neo-Y chromosomes. Proc. Biol. Sci.*

Syren & Luykx. 1977. *Permanent segmental interchange complex in the termite Incisitermes schwarzi. Nature.*

Matsuura K. 2002. *A test of the haplodiploid analogy hypothesis in the termite Reticulitermes speratus (Isoptera: Rhinotermitidae). Annals of the Entomological Society of America.*

Fontana F. 1991. *Multiple reciprocal chromosomal translocations and their role in the evolution of sociality in termites. Ethology Ecology & Evolution*

Additionally, inbreeding through neotenic reproduction (L322) is also avoided in several species through the production of neotenic queens via parthenogenesis.

RESPONSE: We agree with this comment and we now mention that '*in a few termite species, inbreeding is largely avoided through the production of neotenic queens via parthenogenesis, and their interbreeding with the original primary king*' (see Discussion, Lines 388-389).

In addition, we now provide the associated references:

Maatsuura et al. 2009. *Queen succession through asexual reproduction in termites*. *Science*

Vargo et al. 2012. *Asexual queen succession in the subterranean termite *Reticulitermes virginicus**. *Proc. Biol. Sci.*

Hellemans et al. 2019. *Widespread occurrence of asexual reproduction in higher termites of the *Termes* group (Termitidae: Termitinae)*. *BMC evolutionary biology*

Authors show that of 12% of pairs (101 on 831) survived after two weeks. I find that some figures for comparison are generally missing from the main text and would greatly benefit the reader. As such, around 50% of settled pairs survive the first week in *Hodotermes mossambicus* (Darlington et al., 1977. *Insect. Soc.* 24: 353–358). This represents very little, most dispersing alates being lost at an earlier stage mostly through predation: only 0.5% of flying alates would survive the nuptial flight in *Odontotermes assmuthi* (Basalingappa, 1970. *Indian Zool.* 1: 45–50). Although not directly comparable (field observations versus lab ones), I believe these would nicely complement the global picture on colony foundation that the authors depict in their discussion.

RESPONSE: We agree with the comment of the reviewer, which echoes one of the main points of Reviewer 3 that '*choosiness is costly, especially in termite which experience extremely high predation pressure during colony foundation*'. We now discuss our findings comparing previous results of alate survival in the field, highlighting that extremely high predation pressure during colony foundation may act as a strong selective force leading partners to choose the first mate they encounter, regardless of their relatedness or health. We now explicitly mention that '*the influence of other potential selective pressures associated with nuptial flights (e.g., non-mating, predation and resource shortage) may instead lead partners to choose the first mate they encounter, regardless of their relatedness or health. For example, most dispersing alates of the species *Odontotermes assmuthi* are lost through predation, which results in only 0.5% of flying alates surviving the nuptial flight (Basalingappa 1970). In *Hodotermes mossambicus*, even after pairing and digging a first chamber, only about half of the de-alate pairs survive the first week (Darlington et al. 1977). Overall, these results highlight that choosiness is costly in termite, as extremely high predation pressure during colony foundation may act as a strong selective force to quickly find a mate and seek shelter (Nutting 1969, Dial & Vaughan 1987, Korb & Salewski 2000)*' (see Discussion, Lines 289-294).

Another result is the survival of colonies after 15 months, with only 5% (70 of the 1421) of colonies remain. Of these, 49 produced enough workers to be considered in further analyses (L460-461):

24 were inbred and 25 outbred. What about the remaining 21? Please clearly state the breakout for the 70 colonies in section starting at L179.

RESPONSE: We understand the reviewer's concern and now provide the breakout for the 70 colonies. We now mention that '*most of these colonies, 70 out of 85, survived until the end of the experiment (450 days, month 15): 33 were inbred and 37 outbred (see Results, Lines 197-198).*

Minor comments

L53-54: "...scent of related males...in their female relatives.", "related" can be dropped.

RESPONSE: Corrected (see *Introduction*, **Line 56**).

L114: Pairings.

RESPONSE: Corrected (see *Results*, **Line 122**).

L133: susceptibility of/to what?

RESPONSE: To avoid any confusion, we replaced '*susceptibility*' with '*survival*'. The sentence now reads '*the survival of a pairing was associated with the microbial load of the constituent colonies*' (see *Results*, **Lines 139-140**).

L154: "..., level of unweighted..."

RESPONSE: Corrected (see *Results*, **Line 171**).

L160-161: Please correct "...marginally-significantly associated..." to "...marginally associated..."

RESPONSE: Corrected (see *Results*, **Lines 180-181**).

L207: "... a high microbial-load." but not the highest one. Please specify.

RESPONSE: We understand the reviewer's comment as this question is fully addressed in the entire paragraph provided in **Supplementary information S2**, previously available. Briefly, we mentioned that '*interestingly, alates from colony E had the highest mortality rate in our study, despite not showing the highest level of microbial load. First, this result may reflect the presence of highly virulent strains of fungal or bacterial entomopathogens, despite their modest concentration. Second, [this mortality may stem] from internal parasites, such as entomopathogenic nematodes, which are known to occur and to induce high mortality in R. flavipes, despite not being counted in the microbial load. The presence of fungal or bacterial spores on the surface of the infected partner can be removed by grooming between alates (Chouvenc et al. 2009; Yanagawa et al. 2012; Liu et al. 2019), as observed in Z. angusticollis, where grooming between dealates enables them to control low pathogen exposure of a cuticular fungus (Rosengaus et al. 2011). In contrast, the presence of internal parasites is probably harder*

to overcome by immune behaviors of the partner, even when paired with a resistant partner (e.g., pairing AE)'.

We think this information is not of prime relevance to the understanding of the main conclusion of this manuscript, and would be better provided as Supplementary information. However, if the reviewer or editor thinks this information is required to better understand this paper, we will move this paragraph to the main text.

Figure 4: please add on the Figure the colour code of colonies.

RESPONSE: We now add the color code of the colonies on **Figure 4**.

Figure 4a: something seems written in upper left corner?

RESPONSE: We carefully checked the figure, but we cannot find anything written in the upper left corner (as it should be).

Reviewer #2 (Remarks to the Author):

This is an interesting, well designed and thorough study of the effects of inbreeding in a species of termites. Somewhat unexpectedly, foundational success of colonies was not influenced by relatedness between partners, but by their microbial loads. Inbreeding depression does take its toll later in colony development, as inbred workers are more susceptible to entomopathogens, although colony survival was found to be positively correlated with the level of inbreeding (see comment 3). I think the analyses are generally well done, and the paper is well written. However, some of the results are unexpected, which makes it hard to explain for the authors, and to follow the logic for the reader. I also feel that the authors are strongly guided by an expectation of inbreeding depression, which perhaps biases some of their interpretations. I think this is unnecessary, and in fact, the regular occurrence of inbreeding by replacement reproduction may give an alternative expectation, which may be discussed (see comment 8). I have some general comments and some suggestions for improvement:

1. The conclusions that inbreeding takes its toll later in life is an inference based on the finding that inbred workers have higher sensitivity to pathogens. This result is in conflict with the direct observation that the survival of colonies was not negatively affected by inbreeding, in fact on the contrary. Please specify this conclusion, both in the abstract and elsewhere in the manuscript.

RESPONSE: We understand the reviewer's comment; we now address his/her concern in the discussion. We added that *'notably, the higher mortality of inbred offspring contrasts with the absence of an inbreeding effect on the survival of the pairings (parents) over the 15-month study period. This difference may potentially stem from the high pathogen load experimentally used to assess offspring mortality. Similarly, the absence of an effect of inbreeding on the survival of the pairings may also reflect the low and homogeneous pathogen pressure that pairings experienced under lab conditions. Our findings however suggest that, under a more diverse pathogen pressure naturally occurring in the field, the reduced survival of inbred offspring in incipient colonies progressively decreases the proportion of inbred pairings over time'* (see Discussion, **Lines 326-334**).

2. Lines 46-54: Different ways to reduce inbreeding are mentioned. I thought a very obvious one was lacking, the highly synchronised swarming seen in many species.

RESPONSE: We agree with the reviewer and have now integrated this mechanism of inbreeding avoidance. We now mention that *'in some species, the highly synchronized swarming of large number of reproducing individuals may reduce inbreeding by decreasing the chance of mating with a relative'* (see Introduction, **Lines 53-55**).

3. In the results section it is not specified what is meant with inbred and outbred colonies. As far as I can see, inbred colonies are established from sib-alate pairings, and outbred from pairings between alates from different colonies. Please add this information and those results at the start of the results section.

RESPONSE: We understand the reviewer's comment and now provide the information at the beginning of the Results section to help clarify to readers before they encounter the Methods

section. We now explain that *'to investigate the short-term effect of inbreeding on founding success, we set up inbred colonies established from sib-mated pairings and outbred colonies from pairings between alates from different stock colonies for every combination of colonies'* (see Results, Lines 111-113).

4. I also see that relatedness between different source colonies has been estimated using microsatellites, but nowhere is this result being discussed in the result section. Please specify.

RESPONSE: We agree with this comment and thank the reviewers for pointing it out (this point was also mentioned by Reviewer 3). We now mention in the results that *'colonies C and E with the lowest number of surviving alates after 14 days (22 and 8, respectively) also exhibited high levels of relatedness (0.75 and 0.71, respectively), suggesting that these stock colonies were headed by inbred neotenics. In comparison, the degree of relatedness among members of the other colonies (i.e., A, B, D and F) was close to 0.50, indicating they were probably headed by a monogamous pair of outbred primary reproductives (i.e., 0.48, 0.43, 0.52 and 0.54)'* (see Results, Lines 153-159). Although these results do not change our previous conclusions that the relatedness between the partners does not influence pairing survival (inbred vs. outbred pairings), they emphasize the importance of individual inbreeding (inbred vs. outbred alates). This is now highlighted in the Discussion: *'likewise, the primary couple also lacks the benefits of social immunity in the initial stages of colony foundation, suggesting that individual immunity of the founders also plays an important role. In our study, alates from inbred stock colonies (C and E; probably headed by neotenics) suffered high mortality after 14 days, in comparison to alates originating from stock colonies headed by outbred primary reproductives. Hence, although social immunity may allow neotenic inbreeding in populous colonies, those colonies may suffer from producing inbred alates with reduced individual immunity that will not survive long enough to benefit from social immunity that occurs when workers are produced'* (see Discussion, Lines 397-404).

5. Line 190: Here the survival of inbred and outbred workers when challenged with pathogens is described. It took me a while to understand the difference with the first results on initial colony survival. It would help if this experiment is introduced with the details of the entomopathogens used and to describe that individual survival was measured.

RESPONSE: As suggested by the reviewer, we now introduce the paragraph stating that *'in addition to estimating pairing survival, the microbial load and survival of their offspring was also monitored for 14 days following exposure to the entomopathogenic fungus Metarhizium'* (see Results, Lines 219-222). Note that in order to make this point clear throughout the manuscript, we also deleted the ambiguous term *incipient colony survival*. We now consistently use *pairing survival* to describe the mortality of the parents, and *offspring survival* to refer to the workers they produce.

6. Line 220: The title of this section of the discussion is “Avoidance of inbred or unhealthy partner” I think what is meant here is “Avoidance of related or unhealthy partner”. (Inbred partner could be a separate question, not addressed in this study)

RESPONSE: We changed the title of this section as suggested by the reviewer (see *Discussion*, **Line 251**).

7. I found figure 6 not very clear. Especially the lower part: it contains an X and a single Y axis, but three different dependent variables. Perhaps an additional axis on colony size can be specified on the righthand side of the figure, and then the coloured part can be explained.

RESPONSE: We understand the reviewer’s comment. As suggested, we now add an additional axis on the right hand side representing colony size. Also, we now clearly mention that ‘*the red area represents the efficiency of social immunity, which increases with colony size until it is expected to slightly decrease due to inbreeding from neotenic reproduction. The high efficiency of social immunity in large mature colonies releases inbreeding depression, allowing the development of inbred neotenic reproductives without suffering costs associated with pathogen pressure*’. We now also show the cost of inbreeding resulting from the reduced individual immunity of inbred alates during the early stages of colony foundation (mentioned in Rev1. Comment 6). We now illustrate and mention that ‘*the grey area represents the influence of individual immunity, which is high in founding couples and in small incipient colonies. The cost of inbreeding in inbred alates (grey line) is high before incipient colonies become large enough to benefit from social immunity*’ (see *Figure caption*, **Lines 668-673**).

8. Lines 350-351: “In contrast, the common occurrence of inbreeding among neotenics in mature termite colonies suggests a lower level of inbreeding depression.” While this is true, the authors could perhaps add that the low level of inbreeding depression may in fact be a consequence of frequent mating between neotenics, as this will purge deleterious alleles. So in other words, frequent inbreeding makes them resistant to inbreeding.

RESPONSE: We fully agree with this comment and thank the reviewers for pointing it out (this point was also mentioned by Reviewer 3). In the discussion, we now added ‘*the frequent inbreeding in mature termite colonies suggests a reduced level of inbreeding depression. Reduced inbreeding depression may in fact be a consequence of frequent mating between neotenics, as the common occurrence of inbreeding within population is expected to result in a purge of deleterious alleles over time*’ (see *Discussion*, **Lines 374-377**).

In addition, we now provide the associated references:

(Barrett et al. 1991. Effects of a change in the level of inbreeding on the genetic load. *Nature*, 352, 522. Day et al. 2003. The influence of variable rates of inbreeding on fitness, environmental responsiveness, and evolutionary potential. *Evolution*, 57, 1314–1324. Crnokrak et al. 2002. Perspective: Purging the genetic load: A review of the experimental evidence. *Evolution*, 56, 2347–2358. Eyer et al. Inbreeding tolerance as a pre-adapted trait for invasion success in the invasive ant *Brachyponera chinensis*. *Mol. Ecol.* 2018; 27: 4711– 4724).

Reviewer #3 (Remarks to the Author):

The authors investigated the cost of inbreeding in the termite *Reticulitermes flavipes*, paying special attention to the importance of pathogens. Starting with a large number of termite couples set up in the lab either under inbred or no-inbred conditions, their results showed that inbreeding does not seem to have fundamental negative effects on the colonies' success. Inbred colonies grew faster during early colony development but could cope less well during experimental infections with pathogenic *Metarhizium* fungi. The experiments are comprehensive and well done. The authors add with their study new data to a controversial topic. Thus, the study is of relevance for people from the field but also beyond. The latter aspect can be worked out more in a revision. I have some points that can and should be addressed in a revision.

Major comments

Some revision of the text is needful. Some sentences are unclear or illogical (see below). Some precision of terms and concepts are required: Outbreeding, inbreeding depression. Furthermore, some sentences should be more carefully phrased as they are based on results from a single/few species, neglecting the diversity in termites; see below. In addition, I have some statistical and methodological comments that should be considered.

RESPONSE: We have changed the manuscript taking into account all comments for the three reviewers. We hope these changes have improved the manuscript by clarifying some statements, providing additional references highlighting termite diversity, and revising some statistical analyses.

The title should be adapted; given the lifespan of colonies the experiments were not long-term.

RESPONSE: We agree with the reviewer that we investigated colony development during the early stages of colony foundation and we did not track colonies until maturity. We however used alates from mature colonies to study how inbreeding affects alate pairing, offspring production and offspring resistance. Our results therefore provide insights into the cost of inbreeding at various stages of colony development, and reveal that the cost of inbreeding during colony development favors outbred colonies reaching maturity. Although our experiments did not cover the full lifespan of colonies, we however thoroughly discussed our results with previously published findings on mature field colonies. Altogether, we proposed a model for how individual immunity and social immunity change over the life of a colony (from alate pairings to mature colonies, to the production of a new generation of alates by mature colonies). We therefore think that, despite not tracking colonies through to maturity in our experiments, our findings and conclusions cover the full lifespan of colonies.

Although the termite drawings are nice and might work in an oral presentation, they should be removed in the manuscript. They are too prominent and distract from the main content of the figures. Some figures are unnecessarily complicated. Please, simplify (see below).

RESPONSE: We substantially decreased the size of some of the drawings to avoid them from distracting from the main content of the figure. As detailed below, we now provide additional

explanations for the figures, enabling readers unfamiliar with the analyses to more easily understand the graphic presentation of the results (see detailed response below).

Details and Other comments

Methods & Stat.

Relatedness estimates: Please provide confidence intervals for relatedness estimates. Figure S1 requires better explanation. What do you mean with relatedness coefficients were weighted equally? What was the background allele frequency against which relatedness was estimated?

RESPONSE: We now indicate the variance estimates for relatedness values in Supplementary Figure S3. Note that, as expected, the variance estimates are low (ranging from 0.011 to 0.063). We also now provide additional explanation for Figure S1, enabling readers unfamiliar with those analyses to more easily comprehend the results presented graphically. We now explicitly mention that '*dots represent individuals, each dot is colored according to its nest of origin. Individuals from nests belonging to the same colony are expected to cluster together, while individuals from different colonies should segregate across the axes*' (see Figure captions, **Lines 567-569**). The sentence stating that relatedness coefficients were weighted equally was removed from the manuscript.

We agree with the reviewer that explicitly stating the background allele frequency against which relatedness is estimated is important in cases where this information may be ambiguous (*i.e.*, when the dataset contains *at least* two populations with potentially different allelic frequencies). However, our study includes colonies sampled from a single population. It is therefore obvious that the background allele frequency used was the one from this population. We therefore do not see the point of specifying such obvious information.

I.431: Please correct for multiple testing

RESPONSE: We respectfully disagree with the reviewer as this Mann–Whitney U-test was used to compare Euclidean distances between only two groups of individuals (individuals within colonies vs. individuals between colonies). As only two groups were compared, there is no need for correcting for multiple testing.

I. 454-456: Time should be a covariate not a factor, and 'type of pairing' a fixed factor as you manipulated it purposefully (even though you arbitrarily assigned pairs, I guess)

RESPONSE: We now use linear regressions to test for '*the relationship between the numbers of workers and soldiers present in colonies as a function the type of pairing (inbred or outbred), with time tested as a covariable*' (see Methods, **Lines 528-530**).

Note that this change does not modify the conclusion of these tests. The significance of the effect of the type of pairing on the production of workers was $P = 0.000495$; it is now $P = 0.000253$ under the new analysis. Similarly the significance of the effect of the type of pairing on the production of soldiers was $P = 0.000429$; it is now $P = 0.000000516$. Therefore no change was made to the results as we still find that '*the type of pairing significantly affects the number of workers present in colonies over time, with a higher production of workers in inbred colonies ($P <$*

0.001). [...] A similar effect of the type of pairing was found on the number of soldiers over time, with an increased production in inbred colonies ($P < 0.001$)' (see Results, Lines 208-216).

Others

I.28-29: Unclear sentence. Seem illogical, how can a faster growth (something positive) reveal a trade-off.

RESPONSE: We agree with the reviewer and have changed the sentence. It now reads 'we showed faster growth in inbred colonies with low level of microbial load, revealing a potential tradeoff between pathogen defense and offspring production' (see Abstract, Lines 28-30).

I.31-32: unclear sentence. Why 'although'? Change conclusion: I think it is also not correct to say that outbreeding is generally favoured. (i) this only applies with pathogen infection, and (ii) it would mean that inbreeding is disfavoured. Finally, the colonies were not tested up to the stage of maturity (i.e., when the first sexual offspring are produced).

RESPONSE: The sentences mentioned state that '*inbreeding takes its toll later in colony development when offspring from incipient colonies face pathogen pressure. Although the success of a lifetime partnership is initially determined by the partner's health, the cost of inbreeding in incipient colonies favors outbred colonies reaching maturity*' (see Abstract, Lines 30-33). We therefore do not understand this comment, as we explicitly mentioned that (ii) inbreeding is costly and that (i) it particularly applies to pathogen infection.

As mentioned above, our results provide insights into the cost of inbreeding at various stages of colony development. Although our experiments did not cover the full lifespan of colonies, we proposed a model for how individual immunity and social immunity change over the life of a colony (from alate pairings to mature colonies, to the production of a new generation of alates by mature colonies). We therefore think that, despite not tracking colonies through to maturity in our experiments, our findings and conclusions cover the full lifespan of colonies.

I. 44- 54: This section – as well as the discussion – overlooks purging selecting: the removal of slightly deleterious alleles over time when inbreeding commonly occurs. Which is probably the case, given that termite commonly produce neotenic. A fact that is often overlooked in the termite literature.

RESPONSE: We fully agree with this comment, also pointed out by Reviewer 2 (comment 8). As mentioned above, we now integrate in the discussion that '*the frequent inbreeding in mature termite colonies suggests a reduced level of inbreeding depression. Interestingly, reduced inbreeding depression may in fact be a consequence of frequent mating between neotenic, as the common occurrence of inbreeding within population is expected to result in a purge of deleterious alleles over time*' (see Discussion, Lines 374-377)

We now also provide the associated references: (Barrett et al. 1991. Effects of a change in the level of inbreeding on the genetic load. *Nature*, 352, 522. Day et al. 2003. The influence of variable rates of inbreeding on fitness, environmental responsiveness, and evolutionary potential.

Evolution, 57, 1314–1324. Crnokrak et al. 2002. Perspective: Purging the genetic load: A review of the experimental evidence. *Evolution*, 56, 2347–2358. Eyer et al. Inbreeding tolerance as a pre-adapted trait for invasion success in the invasive ant *Brachyponera chinensis*. *Mol. Ecol.* 2018; 27: 4711– 4724).

I.57-58: The opportunity for extra-pair fertilizations largely depends on the taxonomic group/life type in termites: in wood-dwelling dampwood and drywood termites it is very common see e.g. termite review: Korb & Thorne 2017, and work on *Z. nevadensis* by Thorne group (e.g. Thorne et al. 2003), and on *Cryptotermes secundus* by Korb group (Schneider & Korb 2007, Korb & Roux 2021)

RESPONSE: We agree with the reviewer that the opportunity for extra-pair fertilizations largely depends on the taxonomic group/life type in termites. We however prefer not to provide extensive details and comparisons of which species/genus or life type may favor colony fusion in termites. We now mention that colony fusion may allow extra-pair fertilizations in some cases (instead of rare cases). In addition, we now provide the references suggested by the reviewer at the end of this statement (see *Introduction*, *Lines 59-61*).

I. 65: This should be written more carefully because this conclusion is just based on *Zootermopsis*, as a dampwood termite it is exposed to exceptional high pathogen loads, which differs e.g. compared to drywoods.

RESPONSE: We changed the sentence to explicitly mention that '*in incipient colonies, the parents' limited resources are drained by the production and care of the first brood, which is altricial for the two first instars and potentially more susceptible to pathogens than older workers*' (see *Introduction*, **Lines 66-69**).

I.69-71: well both partners are primarily also required for mating as only a minority of species can re-produce parthenogenetically.

RESPONSE: We now mention that '*these behavioral and physiological changes highlight that, in addition of its requirement for mating, the presence of both partners and their mutual compatibility play an important role in influencing the success of incipient colonies*' (see *Introduction*, **Lines 72-75**).

I. 87 and ff.: outbred pairings must be defined. Often 'outbreeding' means less closely related than expected by chance. This is not the case here; here it is rather non-inbred.

RESPONSE: We now define outbreeding ahead in the introduction as the mating of an unrelated queen and king (see *Introduction*, **Lines 58-59**). In addition, we now replace *outbred* with *non-inbred* pairings in the sentence mentioned (see *Introduction*, **Lines 88-92**).

I.98: and in the title: 15 months is not really longterm given the longevity of queens/kings /the colony. Please change throughout.

RESPONSE: As mentioned above, we used alates from mature colonies to study how inbreeding affects alate pairing, offspring production and offspring resistance. We provided insights into the cost of inbreeding at various stages of colony development, which reveal how individual immunity and social immunity change over the life of a colony. We therefore think that, despite not tracking colonies through to maturity in our experiments, our findings and conclusions allow us to determine the long-term costs of inbreeding in termite colonies.

The introduction is missing the important part that choosiness is costly, especially in termite which have experience extremely high predation pressure during colony foundation. This should also be more stressed in the discussion. In the field alates/dealates are just eaten by predators; every extra minute outside means a very high risk to be eaten. This strongly selects against choosiness in termite which is also often neglected in such discussion.

RESPONSE: We agree with the comment of the reviewer, which is also pointed out by Reviewer 1 (see Rev 1-comment 3). As mentioned above, we now state that extremely high predation pressure during colony foundation may act as a strong selective force, leading partners to choose the first mate they encounter, regardless of their relatedness or health. We now explicitly mention that *'our results revealed that the high risk of pairing with a sick partner represents most of the mortality observed during colony foundation, which suggests that pathogen recognition and avoidance should act as a strong selective force [...]. However, the influence of other potential selective pressures associated with nuptial flights (e.g., non-mating, predation and resource shortage) may instead lead partners to choose the first mate they encounter, regardless of their relatedness or health. For example, most dispersing alates of the species *Odontotermes assmuthi* are lost through predation, which results in only 0.5% of flying alates surviving the nuptial flight (Basalingappa 1970). In *Hodotermes mossambicus*, even after pairing and digging a first chamber, only about half of the de-alate pairs survive the first week (Darlington et al. 1977). Overall, these results highlight that choosiness is costly in termite, as extremely high predation pressure during colony foundation may act as a strong selective force to quickly find a mate and seek shelter (Nutting 1969, Dial & Vaughan 1987, Korb & Salewski 2000)' (see Discussion, Lines 281-294).*

Results: As the methods are presented after the Results, more information need to be given in the results so that a reader can understand: e.g. (i) the name of the statistical tests, incl. what was specially tested: random factor, fixed factor..; (ii) explain what Unifrac values are and the difference between weighted and unweighted, (iii) explain highest/cumulative microbial load..

RESPONSE: We understand the reviewer's comment and now provide relevant information on the methods in the results section.

(ii) We now mention in the Results that *'unweighted distances only consider the presence or absence of observed microbes, while weighted values also account for their abundance'* (see Results, Lines 168-169).

(iii) We now mention in the Results that *'the survival of a pairing was associated with the microbial load of the constituent colonies, when considering only the colony of origin with the highest microbial load value ($P = 0.0009$) and when considering the cumulative microbial load level carried by both partners (i.e., the sum of the microbial load across the two colonies of origin)'* (see Results, **Lines 139-142**).

(i) We however think that the sentences in the Results section easily enable the reader to figure out what was specially tested, and there is therefore no need to repeat this information in each sentence (e.g., *No significant difference was observed between the survival of inbred and outbred pairings ($P = 0.212$)* (see Results, **Lines 115-116**). *No significant difference was observed between the survival of inbred and outbred pairings over the course of the experiment ($P = 0.465$)* (see Results, **Lines 199-200**). *This results in similar levels of weighted bacterial differentiation observed within colonies and between different colonies ($P = 0.733$)* (see Results, **Lines 169-171**)).

Similarly, we think that providing the random factor, fixed factor, or the name of the statistical tests is not required to understand the Results section, it is just repetition of information fully available in the Methods section.

I.114: typo: pairings

RESPONSE: Corrected (see Results, **Line 122**)

Figure 2. This figure is unnecessarily complicated and it is hard to see the important results. (a) and (c) can, e.g. be just presented as bar plots. Pie charts not very useful, exchange them for staged bars. (e.g. for c. colonies along the x-axis, pie charts as staged bar. The radar plot is incomprehensible and not explained.

RESPONSE: In our opinion, both pie charts and staged bars are really basic representations of data; both of them are easily understood by all readers. We therefore think that using one or the other is personal preference, but does not hamper the comprehension of the results. For this reason, we prefer to use pie charts. We note that neither of the other two reviewers had an issue with this figure.

We now provide additional explanation for the radar plot (see *Figure Captions, Lines 629-631*), explicitly mentioning that *'radar plot represents the hazard ratio of each inbred and outbred pairings in the first 14 days after colony establishment. Pairings with low hazard ratios (i.e., close to the center) are characterized by low mortality. Pairings with at least one of the partners originating for the stock colony A are colored in blue (B in red; C in orange; D in green; E in purple and F in light blue). Outbred pairings are marked with a circle, while outbred pairings are represented with a square'*.

- Because blue pairings (i.e., pairings with A) are obviously in the center of the graph, the reader can now easily draw the conclusion we mentioned in the results: *'alates from colony A had the highest survival rate [...] Pairings including an alate from A showed good survival overall (low hazard ratio), with the best survival observed for the inbred AA combination'*.

- Because purple pairings (i.e., pairings with E) occupy the outer region of the graph, the conclusion mentioned in the results is also easily drawn: *'notably, the opposite was also observed,*

with alates from colony E having the highest mortality rate. Consequently, pairings including an alate from this colony had low survival, with the lowest survival observed for the inbred pairing EE.

I am wondering about the relatedness values. Figure 3 implies that colony C and colony E are inbred colonies as r is around 0.75, while colony B below 0.5 (fused colony?). Is this correct? This should be discussed.

RESPONSE: We thank the reviewer (and Reviewer 1) for pointing this out. As mentioned above, we now mention in the results that *'the colonies C and E with the lowest number of surviving alates after 14 days (22 and 8, respectively) also exhibited high relatedness (0.75 and 0.71, respectively), suggesting that these stock colonies were headed by inbred neotenics. In comparison, the relatedness among members of the other colonies (i.e., A, B, D and F) was close to 0.50, revealing they were probably headed by a monogamous pair of outbred primary reproductives (i.e., 0.48, 0.43, 0.52 and 0.54)'* (see Results, **Lines 153-159**). Although these results do not change our previous conclusions that the relatedness between the partners does not influence pairing survival (inbred vs outbred pairings), they emphasize the importance of individual inbreeding (inbred vs outbred alates). This is now highlighted in the Discussion, where we mention that *'immune behaviors may therefore be more prevalent and efficient in large groups, accounting for the higher influence of individual immunity (related to individual genetic diversity as determined by inbreeding) in small incipient colonies. [...] Likewise, individual immunity of the founding couple may play an important role in the first days of colony foundation. In our study, inbred alates from colonies headed by neotenics (C and E) exhibit high mortality after 14 days, in comparison to outbred alates originating from colonies headed by outbred primary reproductives. Hence, although social immunity may allow neotenic inbreeding in populous colonies, those colonies may suffer from producing inbred alates with reduced individual immunity that will not survive long enough to benefit from social immunity that occurs when workers are produced'* (see Discussion, **Lines 395-404**).

I.168 add %

RESPONSE: We added that 85 out of the 1421 represent 5.98% of surviving (see Results, **Lines 195-196**)

I.174: Which test was applied?

RESPONSE: We now explicitly mention in the Methods that *'linear regression was performed to determine the relationship between the hazard ratios at 14 days and at 15 months after pairing of each combination of colonies'* (see Methods, **Lines 532-533**).

I.215: Here and elsewhere: Choosiness was not investigated. The experiments cannot tell whether the termites are in nature choosy. Please, adjust.

RESPONSE: We have removed *'choosiness'* from the sentence, which now reads as *'these findings suggest that although the benefit of a lifetime partner is initially influenced by the*

immediate advantage of a healthy partner rather than the long-term potential of fit offspring' (see Discussion, **Lines 244-246**).

We also made a similar change in the Abstract, mentioning that *'the success of a lifetime partnership is initially determined by the partner's health'* (see Abstract, **Lines 31-32**).

I.218: unclear, what does 'restore' mean? Note, at 15 months the colonies are not mature.

RESPONSE: We now mention that *'inbreeding depression during colony development may favor outbred colonies reaching maturity'* (see Abstract, **Lines 32-33**).

As mentioned above, we provided insights into the cost of inbreeding at various stages of colony development, revealing how individual immunity and social immunity change over the life of a colony. Despite not tracking colonies through to maturity in our experiments, our findings and conclusions allow us to discuss the long-term costs of inbreeding in termite colonies.

I.222: susceptible is the wrong term

RESPONSE: We agree with the reviewer and have changed the sentence. It now reads *'detection and avoidance of partners who are either unhealthy or are nestmates potentially occur during nuptial flights'* (see Results, **Lines 253-254**)

I.220 and ff.: The study cannot say whether termites are choosy in nature. The importance of predation which is a major driver in nature should be pointed out more in this section.

RESPONSE: We agree with the comment of the reviewer. As mentioned above, we now integrate that extremely high predation pressure during colony foundation may act as a strong selective force, selecting against choosiness in termites (see detailed answer above) (see Discussion, **Lines 281-294**).

I.231: there is important literature missing. Actually these indirect mechanisms do not seem to be rare. See major important review by Nutting 1969. Females very generally disperse before males.

RESPONSE: We now provide the reference for the literature review of Nutting 1969 for the sentence mentioning that *'inbreeding avoidance can occur through a split sex-ratio between colonies, or differences between the sexes in their dispersal range or in their timing of emergence'* (see Discussion, **Line 261**)

I.232-234: please also read Nutting 1969 for this. What is genetic differentiation between colonies? Here more real population genetic studies should be cited that measured genetic differentiation e.g. work by Kaib & Brandl, Goodisman, Schmidt & Korb.

RESPONSE: We understand the reviewer's comment. The previous sentence stated: *'long-range dispersal is probably the predominant mechanism preventing inbreeding in many species, as alates can disperse hundreds of meters, which leads to genetic differentiation between closely located populations/colonies'*. We have changed the sentence to remove the wrong idea that

alates are good dispersers, promoting high gene flow between populations. We now explicitly mention that '*In termites, the low genetic similarity between neighboring colonies within populations and the fact that alates fly away from their natal colonies suggest that synchronous alate swarming is probably the predominant mechanism preventing inbreeding in many species (Note that alate dispersal is however often insufficient to maintain gene flow between populations)*' (see *Discussion*, Lines **264-268**).

In addition, we now provide the references suggested by the reviewer (*i.e.*, reporting termites species with genetic structure among populations) at the end of this note.

I.239: what is meant with different stages of a colony lifecycle

RESPONSE: We now explicitly define that '*the large variation in the relatedness between partners [occurs] at different stages of the colony lifecycle (i.e., from colony foundation to mature colonies headed by neotenic reproductives in the case of subterranean termites and other lower termites)*' (see *Discussion*, Lines 273-275).

I.248: predation acts even as a stronger selective force which can be seen when dispersal is studied in nature (Nutting 1969, Korb & Schneider 2007, Korb & Salewski 2000).

RESPONSE: As detailed above, we now integrate the suggested references, and those provided by Reviewer 1 to emphasize that high predation pressure during colony foundation may act as a strong selective force against choosiness in termites (see detailed answer above) (see *Discussion*, Lines **281-294**).

I.252-253: see above, links to relevant termite literature should be provided.

RESPONSE: See comment above.

I.258: unclear sentence

RESPONSE: We changed the sentence to explicitly mention that '*the higher productivity of inbred colonies (with low microbial load) may therefore stem from a trade-off in resource investment between pathogen defense and offspring production*' (see *Discussion*, Lines **299-300**).

I.265 and ff.: I agree completely that the colony of origin is an important factor that was not considered in other studies. Yet, equal numbers of colonies in set up are less important. This is taken into account with the statistical analyses, when you include colony of origin as a factor. One could /should even include the interaction between 'Colony of origin' and 'inbred: yes, no' as factors. This will reveal these interactions.

RESPONSE: We now clearly mention that '*most studies investigating differences in survival or productivity between inbred and outbred colonies have not [...] taken into account the colony of origin (potentially testing for an interaction effect with the type of pairing)*' (see *Discussion*, Lines **306-309**).

I.279-281: If you did not find differences in microbial loads between inbred and non-inbred colonies, than you cannot explain this with results that in another study these differences were found.

RESPONSE: We understand the concern of the reviewer and now emphasize the difference between the two studies. This portion of the manuscript now reads as *'our results show that incipient colonies may suffer from inbreeding when facing pathogen pressure, although cuticular microbial loads did not differ between inbred and outbred offspring. In contrast, higher microbial loads were observed in inbred colonies of Z. angusticollis, potentially resulting from reduced grooming or less effective antimicrobials'* (see *Discussion*, **Lines 319-326**)

I.288: reference format.

RESPONSE: Corrected (see *Discussion*, **Line 340**)

I.293-298: important to add 'under conditions of high pathogen load'.

RESPONSE: We agree with the Reviewer and now explicitly mention that *'inbreeding [...] may have an important role in incipient colonies under conditions of high pathogen load'* (see *Discussion*, **Lines 345-347**). Similarly, we also state that the higher mortality of inbred offspring *'may potentially stem from the high pathogen load artificially used to assess offspring mortality'* (see *Discussion*, **Lines 328-329**).

I.324 and ff: unclear. Also in vertebrates, individuals prevent same sex individuals from reproducing breeding.. same in social insects. The removal of reproductives trigger development of same sex reproductives and sometimes foster the development opposite sex reproductives (but only if there is none.. Add References

RESPONSE: We understand the Reviewer's concern and now explicitly state that *'the removal of reproductives triggers development of same sex reproductives and sometimes fosters the development of opposite sex reproductives'*. We now provide the associated reference (Sun et al. 2017. Sex-specific inhibition and stimulation of worker-reproductive transition in a termite *The Science of Nature* 104 (9), 1-8) (see *Discussion*, **Lines 372-373**).

I.303 and ff: this section should include considerations of purging selection (major disadvantages of inbreeding disappear via purging selection) and predation.

RESPONSE: We fully agree with this comment, also pointed out by Reviewer 2 (comment 8). As mentioned above, we now integrate in the discussion that *'the frequent inbreeding in mature termite colonies suggests a reduced level of inbreeding depression. Interestingly, the low level of inbreeding depression may in fact be a consequence of frequent mating between neotenic, as the occurrence of low levels of inbreeding within populations is expected to result in a purge of deleterious alleles over time'* (see *Discussion*, **Lines 374-377**) (Barrett et al. 1991. Effects of a

change in the level of inbreeding on the genetic load. *Nature*, 352, 522. Day et al. 2003. The influence of variable rates of inbreeding on fitness, environmental responsiveness, and evolutionary potential. *Evolution*, 57, 1314–1324. Crnokrak et al. 2002. Perspective: Purging the genetic load: A review of the experimental evidence. *Evolution*, 56, 2347–2358. Eyer et al. Inbreeding tolerance as a pre-adapted trait for invasion success in the invasive ant *Brachyponera chinensis*. *Mol. Ecol.* 2018; 27: 4711– 4724).

As also mentioned above (see Rev 1-comment 3), we now emphasize that extremely high predation pressure during colony foundation may act as a strong selective force, leading partners to choose the first mate they encounter, regardless of their relatedness or health (see *Discussion*, Lines **281-294**).

I.338-339: the reference numbers are assigned correctly. Furthermore, the statement is not correct as currently evidence seem equal strong for both hypotheses.

RESPONSE: We have changed the sentence to mention that ‘*similarly, the development of social immunity in shaping disease resistance in termites (also in social Hymenoptera) is hypothesized (instead of ‘seems’) to occur at the expense of individual immunity, as the evolution of sociality is associated with a reduction in their immune gene repertoire 66-68 (but see 69,70)*’ (see *Discussion*, Lines **407-410**).

I.350: This depends on the exact mode of sex determination in Hymenoptera. As is now known it can be zygoty at specific loci that matter.

RESPONSE: We understand the Reviewer’s comment, however we prefer not to extensively detail the differential impacts of single locus CSD versus multiple locus CSD on inbreeding costs in Hymenoptera. We however change the sentence to make clear that ‘*in social Hymenoptera, due to the extra cost of inbreeding resulting from their haplodiploid sex determination, in which a single founding queen cannot afford the burden of producing up to 50% workless and sterile diploid males*’ (see *Discussion*, **Line 422**).

REVIEWERS' COMMENTS:

Reviewer #1 (Remarks to the Author):

The revised manuscript as well as the rebuttal letter were both available for assessment. I am highly satisfied of the response of authors to my previous comments. In addition, I believe that comments and concerns raised by the two other referees have been reasonably addressed as well.

I agree that my comment on L207 was somewhat confusing and that all information was already available as Supplementary Information S2; maybe this material could be cited there again? Also, I now attach a screenshot indicating the text in upper left corner defined by the PCoA axes in Figure 4a.

Reviewer #2 (Remarks to the Author):

I am happy with the changes you made, and the paper is acceptable now. Congratulations with this interesting study!

Reviewer #3 (Remarks to the Author):

Thanks for the revision. I am now happy to accept the manuscript.

Reviewer #1:

The revised manuscript as well as the rebuttal letter were both available for assessment. I am highly satisfied of the response of authors to my previous comments. In addition, I believe that comments and concerns raised by the two other referees have been reasonably addressed as well.

RESPONSE: We thank the reviewer for his/her enthusiasm for our previous responses.

I agree that my comment on L207 was somewhat confusing and that all information was already available as Supplementary Information S2; maybe this material could be cited there again?

RESPONSE: We thank the reviewer for clarifying his/her previous comment. We now also cite the Supplementary Information (now called 'Supplementary Note 1') at the end of the sentence mentioned by the reviewer.

Also, I now attach a screenshot indicating the text in upper left corner defined by the PCoA axes in Figure 4a.

RESPONSE: We thank the reviewer for providing the screenshot to his/her previous comment. We were able to find and deleted the text in the upper left corner (a white font text on a white background).

Reviewer #2:

I am happy with the changes you made, and the paper is acceptable now. Congratulations with this interesting study!

RESPONSE: We thank the reviewer for his/her enthusiasm for our previous responses.

Reviewer #3:

Thanks for the revision. I am now happy to accept the manuscript.

RESPONSE: We thank the reviewer for his/her enthusiasm for our previous responses.